# Contrastive Counterfactual Generation for Imperceptible Adversarial Attacks

## Abstract

Imperceptible adversarial attacks aim to mislead deep neural networks by adding signal-domain perturbations that induce misclassification while remaining visually indistinguishable from the original signal. Existing methods rely on untargeted loss maximisation, producing perturbations poorly aligned with decision boundaries and providing limited control over locality and perceptual cost. To address these limitations, we propose **Contrastive Counterfactual Generation** (CoCoGen), an adversarial attack framework that operates under a composite threat model combining an $\ell_\infty$ magnitude budget, a sparsity budget with $k$ selected adaptively, and a high-frequency spectral constraint. Within this composite feasible set, CoCoGen formulates perturbation synthesis as a constrained optimisation problem that explicitly targets the nearest decision boundary by minimising the *contrastive counterfactual margin*. Perturbations are localised via gradient-based Top-$k$ spatial projection and confined to the high-frequency subspace using a Fourier-domain projection operator, leveraging reduced human sensitivity to high spatial frequencies. The objective is optimised using masked gradient descent with momentum, while an adaptive sparsity grid search identifies minimal feasible signal support. Experiments across multiple architectures show that CoCoGen achieves 100% Attack Success Rate, compared to 80–99% for most prior methods (up to 100% for one baseline), while maintaining a MUSIQ score of 61–63 (vs. 36–55), outperforming prior methods in both attack efficacy and visual quality. We further validate CoCoGen with a human perceptual study showing its perturbations are rarely detected or correctly localised by naive observers, and demonstrate consistent gains across an additional dataset and architecture family; to our knowledge, no prior sparse or frequency-constrained attack has been jointly validated with human-perceptual evidence and cross-dataset generalisation.

## 1 Introduction

Deep Neural Networks (DNNs) are vulnerable to adversarial attacks Goodfellow et al. (2015), where carefully crafted signal-domain perturbations added to input signals induce incorrect predictions, posing serious security risks in real-world applications Yuan et al. (2019). Beyond exposing vulnerabilities, adversarial attacks are instrumental in evaluating model robustness and motivating defence strategies Lee & Kim (2023); Singh et al. (2024); Luo et al. (2023); Tramèr et al. (2018); Salman et al. (2020); Luo et al. (2021); Cohen et al. (2019). Many existing attacks Madry et al. (2018); Wei et al. (2023) maximise success under loose signal-energy budgets (*e.g.* $\ell_\infty$ or $\ell_2$ norms), often producing perceptible signal artefacts detectable by the human visual system (HVS) Sharif et al. (2018). This has motivated growing interest in *imperceptible* adversarial attacks Carlini & Wagner (2017); Luo et al. (2018); Zhao et al. (2020); Laidlaw et al. (2021); Duan et al. (2021); Chen et al. (2023c); Jia et al. (2022), which seek to maintain attack efficacy while preserving the perceptual fidelity of the adversarial signal.

Existing imperceptible attacks are broadly categorised into perturbation-constrained and unrestricted methods. Perturbation-based approaches exploit perceptual signal properties such as colour sensitivity Zhao et al. (2020), texture complexity Fang et al. (2026), and frequency characteristics Jia et al. (2022); Luo et al. (2022) to conceal adversarial signal components in perceptually insensitive regions. Unrestricted

attacks modify semantic attributes via generative or diffusion-based models Song et al. (2018); Rombach et al. (2022); Chen et al. (2023b); Xue et al. (2023); Chen et al. (2023a; 2024), but often introduce unnatural signal distortions, particularly for complex scenes. More fundamentally, most methods rely on *untargeted loss maximisation*, showing no ⸻ ary nor to concentrate signal energy on the compor⸻

To address these limitations, we propose **Co**ntrastive **Co**unterfactual **Gen**eration CoCoGen, an adversarial attack that minimises the contrastive counterfactual margin Moosavi-Dezfooli et al. (2016); Carlini & Wagner (2017), focusing perturbations on the most competitive incorrect class. Specifically, CoCoGen integrates Top-$k$ spatial projection Papernot et al. (2016); Modas et al. (2019); Croce et al. (2022), Fourier high-frequency constraints Luo et al. (2022), and masked momentum iterative updates Dong et al. (2018) with adaptive sparsity and spectral threshold search to produce imperceptible perturbations. Unlike prior methods that typically fix either the perturbation support or the frequency-domain constraint, CoCoGen jointly searches for both the sparsity level $k$ and the spectral threshold while enforcing perceptual-quality constraints, operating under the composite feasible. CoCoGen achieves **100%** ASR across all models, outperforming prior methods (mostly 80-99%) with lower LPIPS ($\approx 0.01$) and higher MUSIQ (61-63) under this more restrictive threat model.

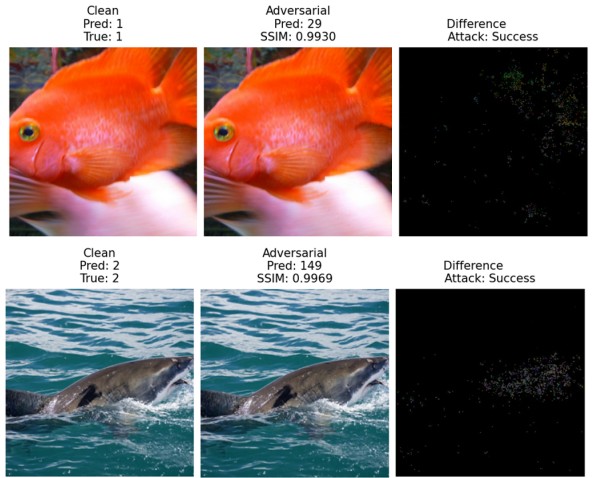

Figure 1: **Imperceptibility of CoCoGen.** Clean signal with labels (col. 1), adversarial signal (col. 2), and contrast-enhanced perturbation map (col. 3) ($\times 10$ for visibility). The perturbations are sparse and confined to high-frequency, thus imperceptible.

The contributions of this work are as follows:

- We propose CoCoGen, an adversarial attack that minimises the contrastive counterfactual margin, focusing perturbations on the most competitive incorrect class under explicit spatial and spectral constraints. Our core contribution is the *joint, adaptive* integration of Top-$k$ spatial masking, Fourier high-frequency projection, and masked momentum updates within a single constrained optimisation framework that searches over sparsity and spectral thresholds simultaneously under perceptual-quality constraints. This is unlike prior methods, which apply these components with a fixed budget in isolation.

- CoCoGen achieves **100%** ASR across four diverse architectures, outperforming prior methods (mostly 80–99%) while maintaining lower LPIPS ($\approx 0.01$) and higher MUSIQ (61–63); against dedicated sparse attacks with comparable or smaller pixel budgets, CoCoGen is the only method to reach 100% ASR on every architecture, showing that sparsity alone does not explain its effectiveness.

- We provide qualitative comparisons, a human perceptual study, and results on an additional dataset that together corroborate our quantitative imperceptibility metrics, showing that CoCoGen's perturbations remain difficult for naive human observers to detect or localise while generalising beyond a single benchmark; to our knowledge, no prior sparse or frequency-constrained attack has been jointly validated with human-perceptual evidence and cross-dataset generalisation.

## 2 Proposed Method: CoCoGen

### 2.1 Problem Formulation

Let $\boldsymbol{x} \in \mathbb{R}^N$ denote a vectorised image, where $N = H \cdot W \cdot C$ is the total number of signal entries obtained by flattening the spatial dimensions $H$ (height) and $W$ (width) over $C$ colour channels (note that this explanation is simplified; for convolutional neural networks, the representation differs). A neural classifier $f : \mathbb{R}^N \to \mathbb{R}^{|\mathcal{Y}|}$

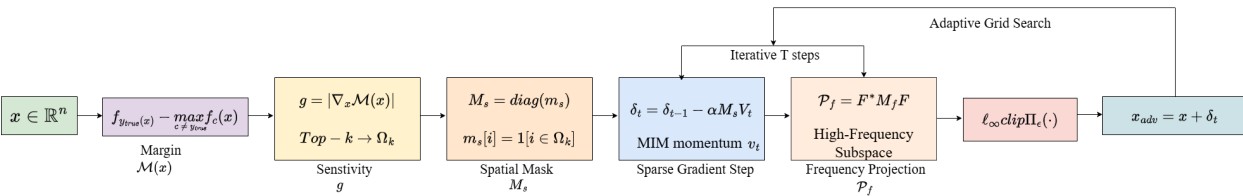

Figure 2: **CoCoGen attack pipeline.** The input signal $\boldsymbol{x}$ is processed through four stages: contrastive margin computation, gradient-based Top-$k$ spatial projection $\boldsymbol{M}_s$, Fourier-domain frequency projection $\mathcal{P}_f$, and $\ell_\infty$ clipping $\Pi_\epsilon$, iterated over $T$ steps with an adaptive sparsity grid search over $k^\star$ to yield the adversarial signal $\boldsymbol{x}_{\mathrm{adv}}$.

produces a logit vector $f(\boldsymbol{x})$, with $f_c(\boldsymbol{x}) \in \mathbb{R}$ denoting the score for class $c \in \mathcal{Y}$. We write $y_{\mathrm{true}} \in \mathcal{Y}$ for the ground-truth label of $\boldsymbol{x}$. We seek a perturbation $\boldsymbol{\delta} \in \mathbb{R}^N$ such that the adversarial example

$$\boldsymbol{x}_{\mathrm{adv}} = \boldsymbol{x} + \boldsymbol{\delta} \tag{1}$$

is misclassified by $f$ while remaining visually indistinguishable from $\boldsymbol{x}$. equation 1 is the standard additive perturbation model Goodfellow et al. (2015). Imperceptibility is enforced via an $\ell_\infty$ budget $\|\boldsymbol{\delta}\|_\infty \leq \epsilon$, where $\epsilon > 0$ is a user-specified tolerance (typically $\epsilon \in [4/255, 16/255]$ for 8-bit images; scaled accordingly for other bit depths).

**Composite Threat Model.** Unlike standard $\ell_\infty$ attacks such as PGD Madry et al. (2018) and C&W Carlini & Wagner (2017), which operate over the unconstrained $\ell_\infty$ ball $\mathcal{B}_\epsilon = \{\boldsymbol{\delta} : \|\boldsymbol{\delta}\|_\infty \leq \epsilon\}$, CoCoGen operates under a strictly more restrictive composite feasible set

$$\mathcal{F} = \big\{\boldsymbol{\delta} \,:\, \|\boldsymbol{\delta}\|_\infty \leq \epsilon,\ \|\boldsymbol{\delta}\|_0 \leq k,\ \boldsymbol{\delta} \in \mathrm{Im}(\mathcal{P}_f)\big\}, \tag{2}$$

which simultaneously enforces a magnitude constraint ($\ell_\infty$ budget $\epsilon$), a sparsity constraint ($\ell_0$ budget $k$, selected adaptively via Eq. 19), and a spectral constraint (support restricted to the high-frequency Fourier subspace $\mathrm{Im}(\mathcal{P}_f)$, defined in equation 13. Since $\mathcal{F} \subsetneq \mathcal{B}_\epsilon$ strictly whenever $k < N$ or $\boldsymbol{M}_f \neq \boldsymbol{I}$ (Proposition 2(ii)), comparisons between CoCoGen and methods operating under the less restrictive threat model $\mathcal{B}_\epsilon$ should be interpreted as evaluating what is achievable under additional perceptual constraints, rather than as direct superiority claims under identical attack conditions.

## 2.2 Contrastive Counterfactual Margin

The margin formulation in our method builds on the logit-difference objective used in DeepFool Moosavi-Dezfooli et al. (2016) and C&W Carlini & Wagner (2017); our contribution lies not in the margin itself but in combining counterfactual-margin optimisation with adaptive sparse support selection and frequency-aware perturbation generation, as described in Sections 2.3-2.6.

Naïvely minimising a cross-entropy loss diffuses perturbation energy across all $N$ signal components, producing detectable artefacts. Instead, we target the *contrastive counterfactual margin* $\mathcal{M}(\cdot)$:

$$\mathcal{M}(\boldsymbol{x}) = f_{y_{\mathrm{true}}}(\boldsymbol{x}) - \max_{c \neq y_{\mathrm{true}}} f_c(\boldsymbol{x}), \tag{3}$$

where $f_{y_{\mathrm{true}}}(\boldsymbol{x})$ is the logit of the correct class and $\max_{c \neq y_{\mathrm{true}}} f_c(\boldsymbol{x})$ is the highest competing logit. The margin in equation 3 is positive for a correctly classified input and negative once misclassification is achieved. The optimisation problem is therefore

$$\min_{\boldsymbol{\delta}}\ \mathcal{M}(\boldsymbol{x} + \boldsymbol{\delta}), \qquad \text{s.t.} \quad \|\boldsymbol{\delta}\|_\infty \leq \epsilon, \tag{4}$$

where the constraint $\|\boldsymbol{\delta}\|_\infty \leq \epsilon$ bounds the per-entry magnitude of the perturbation.

## 2.3 Spatial Sparsity via a Diagonal Projection Operator

To localise the perturbation to the most decision-relevant signal entries, we compute the element-wise absolute gradient of $\mathcal{M}$ with respect to the input:

$$\boldsymbol{g} \;=\; \left|\nabla_{\boldsymbol{x}}\,\mathcal{M}(\boldsymbol{x})\right| \;\in\; \mathbb{R}_{\geq 0}^{N}, \tag{5}$$

where $|\cdot|$ is applied element-wise and $\boldsymbol{g}[i]$ quantifies the sensitivity of the margin to the $i$-th pixel. From $\boldsymbol{g}$ in equation 5 we identify the index set

$$\Omega_k \;=\; \text{Top-}k(\boldsymbol{g}) \;=\; \{i_1, i_2, \ldots, i_k\}, \tag{6}$$

where $\boldsymbol{g}[i_1] \geq \boldsymbol{g}[i_2] \geq \cdots \geq \boldsymbol{g}[i_k]$, and $k \in \{1, \ldots, N\}$ is the sparsity budget. Using $\Omega_k$ from equation 6, we form a binary mask vector $\boldsymbol{m}_s \in \{0,1\}^N$ with

$$\boldsymbol{m}_s[i] \;=\; \mathbb{1}[i \in \Omega_k], \qquad i = 1, \ldots, N, \tag{7}$$

where $\mathbb{1}[\cdot]$ is the indicator function. The mask in equation 7 is then lifted to a diagonal *spatial projection matrix*

$$\boldsymbol{M}_s \;=\; \text{diag}(\boldsymbol{m}_s) \;\in\; \{0,1\}^{N \times N}, \tag{8}$$

which satisfies $\boldsymbol{M}_s^2 = \boldsymbol{M}_s$ (idempotent projector) and $\boldsymbol{M}_s\boldsymbol{u}$ zeroes all entries of $\boldsymbol{u}$ outside $\Omega_k$.

## 2.4 Frequency Subspace Projection

Spatial sparsity alone does not guarantee imperceptibility, since low-frequency perturbations are highly visible to the human visual system (HVS) Wang et al. (2004). We therefore further restrict $\boldsymbol{\delta}$ to the *high-frequency subspace* of $\mathbb{R}^N$ using the Discrete Fourier Transform (DFT).

The DFT is applied independently per colour channel on the 2D spatial grid $H \times W$. Concretely, for channel $c \in \{1, \ldots, C\}$, let $\boldsymbol{F}_{HW} \in \mathbb{C}^{HW \times HW}$ be the unitary 2D DFT matrix (equivalently $\boldsymbol{F}_H \otimes \boldsymbol{F}_W$, the Kronecker product of the 1D DFT matrices along each spatial axis), defined entry-wise as

$$[\boldsymbol{F}_{HW}]_{p,q} \;=\; \frac{1}{\sqrt{HW}} \exp\left(-2\pi\,\mathrm{i}\left(\frac{p_h\,q_h}{H} + \frac{p_w\,q_w}{W}\right)\right), \tag{9}$$

where $p = (p_h, p_w)$ and $q = (q_h, q_w)$ index the 2D spatial and frequency grids respectively, and $\mathrm{i} = \sqrt{-1}$. The full DFT matrix acting on the vectorised $C$-channel signal is

$$\boldsymbol{F} \;=\; \boldsymbol{I}_C \otimes \boldsymbol{F}_{HW} \;\in\; \mathbb{C}^{N \times N}, \qquad N = H \cdot W \cdot C, \tag{10}$$

which applies $\boldsymbol{F}_{HW}$ independently to each channel and satisfies $\boldsymbol{F}\boldsymbol{F}^* = \boldsymbol{I}_N$.

The high-frequency binary mask $\boldsymbol{m}_f \in \{0,1\}^N$ is constructed as follows. Each vectorised index $i \in \{1, \ldots, N\}$ maps to a triple $(c_i, u_i, v_i)$ where $c_i \in \{1, \ldots, C\}$ is the channel index and $(u_i, v_i) \in \{0, \ldots, H-1\} \times \{0, \ldots, W-1\}$ is the 2D frequency coordinate. The mask retains bins whose radial frequency exceeds a threshold $\tau_{\text{freq}} \geq 0$, selected by adaptive search:

$$\boldsymbol{m}_f[i] \;=\; \mathbb{1}\left[\sqrt{u_i^2 + v_i^2} > \tau_{\text{freq}}\right], \qquad i = 1, \ldots, N, \tag{11}$$

where the threshold is applied identically across all $C$ channels. The corresponding diagonal frequency-selection matrix is

$$\boldsymbol{M}_f \;=\; \text{diag}(\boldsymbol{m}_f) \;\in\; \{0,1\}^{N \times N}. \tag{12}$$

The frequency projection operator is then

$$\mathcal{P}_f \;=\; \boldsymbol{F}^*\boldsymbol{M}_f\boldsymbol{F} \;\in\; \mathbb{C}^{N \times N}, \tag{13}$$

where $\boldsymbol{F}$ maps the signal to the frequency domain per channel, $\boldsymbol{M}_f$ retains only high-frequency components, and $\boldsymbol{F}^*$ maps back to the signal domain. One can verify that $\mathcal{P}_f^2 = \mathcal{P}_f$ (idempotent) and $\mathcal{P}_f^* = \mathcal{P}_f$ (self-adjoint), so $\mathcal{P}_f$ is an orthogonal projector.

While SSAH Luo et al. (2022) also restricts perturbations to the high-frequency Fourier subspace, it does so with a fixed frequency threshold determined a priori. COCOGEN differs in two key respects: (i) the frequency threshold $\tau_{\text{freq}}$ is selected adaptively via the grid search of Section 2.6 (equation 19) rather than fixed before optimisation; and (ii) the sparse spatial support $\Omega_k$ (equation 6) is jointly optimised with the spectral constraint, so both the perturbation support and the frequency subspace are determined by the data rather than set as hyperparameters. This joint adaptivity is the primary spectral novelty of COCOGEN relative to SSAH.

## 2.5 Masked Momentum Iterative Update

We optimise equation 4 using a masked extension of the Momentum Iterative Method (MIM) Dong et al. (2018). Let $\boldsymbol{\delta}_0 = \boldsymbol{0}$ and $\boldsymbol{v}_0 = \boldsymbol{0}$ denote the initial perturbation and momentum buffer, respectively, and let $\alpha > 0$ be the step size. At iteration $t \in \{1, \ldots, T\}$ the following three operations are applied in sequence.

**Momentum accumulation.** The gradient of the margin $\mathcal{M}$ (defined in equation 3) with respect to the current adversarial example is $\ell_1$-normalised and accumulated into the momentum buffer:

$$\boldsymbol{v}_t \; = \; \boldsymbol{v}_{t-1} \; + \; \frac{\nabla_{\boldsymbol{x}}\,\mathcal{M}(\boldsymbol{x} + \boldsymbol{\delta}_{t-1})}{\left\| \nabla_{\boldsymbol{x}}\,\mathcal{M}(\boldsymbol{x} + \boldsymbol{\delta}_{t-1}) \right\|_1}, \tag{14}$$

where $\| \cdot \|_1 = \sum_i |\cdot_i|$ normalises the update to unit scale across iterations.

**Sparse gradient step.** The spatial projection matrix $\boldsymbol{M}_s$ from equation 8 restricts the update to the $k$ decision-critical entries:

$$\tilde{\boldsymbol{\delta}}_t \; = \; \boldsymbol{\delta}_{t-1} \; - \; \alpha\, \boldsymbol{M}_s\, \boldsymbol{v}_t, \tag{15}$$

so that only the components in $\Omega_k$ (equation 6) receive a non-zero gradient signal.

**Composite projection.** The intermediate perturbation $\tilde{\boldsymbol{\delta}}_t$ from equation 15 is first projected onto the high-frequency subspace via $\mathcal{P}_f$ (equation 13), and then clipped to the $\ell_\infty$ ball $\mathcal{B}_\epsilon = \{\boldsymbol{u} : \|\boldsymbol{u}\|_\infty \leq \epsilon\}$ (the $\ell_\infty$ ball $\mathcal{B}_\epsilon = \{\boldsymbol{u} : \|\boldsymbol{u}\|_\infty \leq \epsilon\}$:

$$\boldsymbol{\delta}_t \; = \; \Pi_\epsilon\big(\mathcal{P}_f\, \tilde{\boldsymbol{\delta}}_t\big), \tag{16}$$

where the $\ell_\infty$ projection is $\Pi_\epsilon(\boldsymbol{u}) = \text{clip}(\boldsymbol{u}, -\epsilon, \epsilon)$, applied element-wise. Substituting equation 15 into equation 16 produces the *single closed-form iteration*:

$$\boldsymbol{\delta}_t \; = \; \Pi_\epsilon\Big(\boldsymbol{F}^*\boldsymbol{M}_f\boldsymbol{F}\big(\boldsymbol{\delta}_{t-1} - \alpha\, \boldsymbol{M}_s\, \boldsymbol{v}_t\big)\Big), \tag{17}$$

which involves three closed-form linear operators $(\boldsymbol{M}_s,\, \boldsymbol{F}/\boldsymbol{F}^*,\, \boldsymbol{M}_f)$ plus an element-wise clip, and requires no iterative sub-solver.

## 2.6 Adaptive Sparsity Search

The optimal sparsity budget $k$ is unknown *a priori*: a very small $k$ may fail to achieve misclassification, while a large $k$ wastes perceptual budget. We therefore perform a monotone search over a pre-specified candidate set $\mathcal{K} = \{k_1 < k_2 < \cdots < k_L\} \subset \{1, \ldots, N\}$. For each $k \in \mathcal{K}$, the mask $\boldsymbol{m}_s$ in equation 7 is constructed with $|\Omega_k| = k$ active entries, and $T$ iterations of equation 17 are executed to yield $\boldsymbol{x}_{\text{adv}}^{(k)} = \boldsymbol{x} + \boldsymbol{\delta}_T^{(k)}$ (cf. equation 1). A candidate $k$ is feasible only if it simultaneously satisfies the misclassification condition and two perceptual quality thresholds:

$$\frac{1}{B}\sum_{j=1}^{B}\text{SSIM}\Big(\boldsymbol{x}_{\text{adv}}^{(j,k)},\, \boldsymbol{x}^{(j)}\Big) \; \geq \; \tau_s, \qquad \text{FID}\Big(\big\{\boldsymbol{x}_{\text{adv}}^{(j,k)}\big\}_{j=1}^{B},\, \big\{\boldsymbol{x}^{(j)}\big\}_{j=1}^{B}\Big) \; \leq \; \tau_f, \tag{18}$$

where $B$ denotes the number of images used to evaluate both the perceptual feasibility criterion equation 18 and the misclassification condition in equation 19; in our experiments the full 1,000-image ImageNet subset described in Section 4.1 is used (i.e., $B = 1,000$), so $k^\star$ is the smallest sparsity level achieving 100% ASR across all $B$ images while satisfying the perceptual gates $\tau_s = 0.95$ and $\tau_f = 20$. The value of $k$ is selected independently for each target architecture. Besides, $\tau_s \in (0, 1]$ is the minimum acceptable mean Structural Similarity Index Wang et al. (2004) across the batch, and $\tau_f \geq 0$ is the maximum tolerated Fréchet Inception Distance Heusel et al. (2017) computed over the full evaluation batch. The constraint in equation 18 gates candidates by both local fidelity (SSIM, averaged per image) and distributional realism (FID, computed over the batch).

The optimal sparsity level is the smallest feasible candidate:

$$k^\star = \min\left\{k \in \mathcal{K} \ \Big| \ \underbrace{\frac{1}{B}\sum_{j=1}^{B} \mathbb{1}\left[\operatorname*{argmax}_c f_c\left(\boldsymbol{x}_{\text{adv}}^{(j,k)}\right) \neq y_{\text{true}}^{(j)}\right] = 1}_{\text{all } B \text{ images misclassified}} \ \wedge \ \underbrace{equation\ 18\ \text{holds}}_{\text{perceptually valid}}\right\}, \qquad (19)$$

where $\wedge$ denotes logical conjunction. The final adversarial examples are $\{\boldsymbol{x}_{\text{adv}}^{(j,k^\star)}\}_{j=1}^{B}$ from equation 1. If no $k \in \mathcal{K}$ satisfies equation 19, the attack is declared unsuccessful for this batch.

## 2.7 Complexity Analysis

Each iteration of equation 17 requires: (i) one forward–backward pass through $f$ for the gradient in equation 14, costing $\mathcal{O}(\Phi(f))$ where $\Phi(f)$ denotes the number of floating-point operations (FLOPs) in a single forward pass of $f$; (ii) two DFTs ($\mathcal{O}(N \log N)$) from equation 13; and (iii) two diagonal matrix multiplications ($\mathcal{O}(N)$) from equation 8 and equation 12. The dominant cost is the backward pass, which by standard autodifferentiation requires at most $\mathcal{O}(\Phi(f))$ operations; the same asymptotic order as the forward pass. Over $T$ iterations and $|\mathcal{K}|$ sparsity candidates, the total cost scales as $\mathcal{O}(|\mathcal{K}| \cdot T \cdot \Phi(f))$. Since the $|\mathcal{K}|$ sub-problems are independent, the search over equation 19 is embarrassingly parallel (one can run them parallely on different GPUs and CPUs).

# 3 Theoretical Analysis

We establish four formal results that justify the design of CoCoGen: (i) the contrastive counterfactual margin is a signed distance surrogate to the nearest decision boundary (Theorem 1); (ii) the composite signal projection $\boldsymbol{M}_s\mathcal{P}_f$ is a well-defined bounded linear operator with controlled spectral norm (Proposition 1); and (iii) CoCoGen strictly generalises the C&W Carlini & Wagner (2017) objective while producing a geometrically distinct solution path (Proposition 2).

## 3.1 Contrastive Margin as a Decision-Boundary Surrogate

We first formalise the relationship between the contrastive counterfactual margin $\mathcal{M}(\boldsymbol{x})$ defined in equation 3 and the signed distance of $\boldsymbol{x}$ to the nearest decision boundary of $f$.

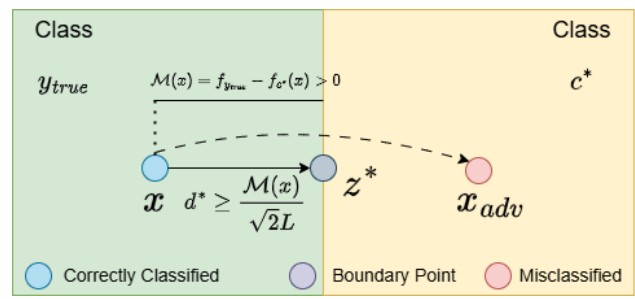

Figure 3: **Decision boundary geometry of CoCoGen.** The contrastive counterfactual margin $\mathcal{M}(\boldsymbol{x}) = f_{y_{\text{true}}}(\boldsymbol{x}) - f_{c^*}(\boldsymbol{x}) > 0$ provides a lower bound on the signed distance $d^*(\boldsymbol{x}) \geq \mathcal{M}(\boldsymbol{x})/(2L)$ to the nearest decision boundary $\mathcal{D}_{y_{\text{true}}, c^*}$. Minimising $\mathcal{M}$ under the signal constraints $\boldsymbol{M}_s$ and $\mathcal{P}_f$ guides $\boldsymbol{\delta}$ toward $z^*$, crossing the boundary with $\mathcal{M}(\boldsymbol{x} + \boldsymbol{\delta}) \leq 0$ using minimal signal energy.

**Assumption 1 (Lipschitz classifier)** *The classifier $f : \mathbb{R}^N \to \mathbb{R}^{|\mathcal{Y}|}$ is $L$-Lipschitz continuous with respect to the $\ell_2$ norm,*

$$\|f(\boldsymbol{x}) - f(\boldsymbol{x}')\|_2 \leq L \|\boldsymbol{x} - \boldsymbol{x}'\|_2, \qquad \forall \boldsymbol{x}, \boldsymbol{x}' \in \mathbb{R}^N, \qquad (20)$$

where $L > 0$ is the global Lipschitz constant of $f$.

**Assumption 2 (Unique runner-up class)** *At the input $\boldsymbol{x}$ with true label $y_{\text{true}}$, the most competitive incorrect class $c^* = \operatorname{argmax}_{c \neq y_{\text{true}}} f_c(\boldsymbol{x})$ is unique, i.e., $f_{c^*}(\boldsymbol{x}) > f_c(\boldsymbol{x})$ for all $c \notin \{y_{\text{true}}, c^*\}$.*

Under Assumption 2, the margin in equation 3 reduces:

$$\mathcal{M}(\boldsymbol{x}) = f_{y_{\text{true}}}(\boldsymbol{x}) - f_{c^*}(\boldsymbol{x}). \tag{21}$$

Let $\mathcal{D}_{y,c} = \{\boldsymbol{z} \in \mathbb{R}^N : f_y(\boldsymbol{z}) = f_c(\boldsymbol{z})\}$ denote the decision boundary between classes $y$ and $c$, and define the signed distance from $\boldsymbol{x}$ to the nearest decision boundary as

$$d^*(\boldsymbol{x}) = \min_{c \neq y_{\text{true}}} \inf_{\boldsymbol{z} \in \mathcal{D}_{y_{\text{true}},c}} \|\boldsymbol{x} - \boldsymbol{z}\|_2. \tag{22}$$

**Theorem 1 (Margin–boundary relationship)** *Under Assumptions 1 and 2, for any $\boldsymbol{x} \in \mathbb{R}^N$ correctly classified by $f$ (refer to Fig. 3):*

(i) **Lower bound:** $d^*(\boldsymbol{x}) \geq \dfrac{\mathcal{M}(\boldsymbol{x})}{2L}$.

(ii) **Monotonicity:** *If perturbation $\boldsymbol{\delta}$ satisfies $\mathcal{M}(\boldsymbol{x} + \boldsymbol{\delta}) < \mathcal{M}(\boldsymbol{x})$, then $d^*(\boldsymbol{x} + \boldsymbol{\delta}) \leq d^*(\boldsymbol{x}) - \dfrac{\Delta}{2L} + \|\boldsymbol{\delta}\|_2$, where $\Delta := \mathcal{M}(\boldsymbol{x}) - \mathcal{M}(\boldsymbol{x} + \boldsymbol{\delta}) > 0$.*

(iii) **Crossing condition:** *$\mathcal{M}(\boldsymbol{x} + \boldsymbol{\delta}) \leq 0$ implies that $\boldsymbol{x} + \boldsymbol{\delta}$ lies on or beyond the decision boundary $\mathcal{D}_{y_{\text{true}},c^*}$.*

*Proof.* **(i) Lower bound.** Let $\boldsymbol{z}^* \in \mathcal{D}_{y_{\text{true}},c^*}$ be a minimiser of equation 22 for class $c^*$. By definition of $\mathcal{D}_{y_{\text{true}},c^*}$, $f_{y_{\text{true}}}(\boldsymbol{z}^*) = f_{c^*}(\boldsymbol{z}^*)$, so

$$\mathcal{M}(\boldsymbol{z}^*) = f_{y_{\text{true}}}(\boldsymbol{z}^*) - f_{c^*}(\boldsymbol{z}^*) = 0. \tag{23}$$

Define $h(\boldsymbol{x}) := f_{y_{\text{true}}}(\boldsymbol{x}) - f_{c^*}(\boldsymbol{x})$, so $\mathcal{M}(\boldsymbol{x}) = h(\boldsymbol{x})$. By the triangle inequality and the $L$-Lipschitz condition on $f$ in equation 20,

$$\begin{aligned} |h(\boldsymbol{x}) - h(\boldsymbol{x}')| &\leq |f_{y_{\text{true}}}(\boldsymbol{x}) - f_{y_{\text{true}}}(\boldsymbol{x}')| + |f_{c^*}(\boldsymbol{x}) - f_{c^*}(\boldsymbol{x}')| \\ &\leq L\|\boldsymbol{x} - \boldsymbol{x}'\|_2 + L\|\boldsymbol{x} - \boldsymbol{x}'\|_2 = 2L\|\boldsymbol{x} - \boldsymbol{x}'\|_2, \end{aligned} \tag{24}$$

so $h$ is $2L$-Lipschitz. Applying equation 24 between $\boldsymbol{x}$ and $\boldsymbol{z}^*$, and using $h(\boldsymbol{z}^*) = 0$:

$$\mathcal{M}(\boldsymbol{x}) = |h(\boldsymbol{x}) - h(\boldsymbol{z}^*)| \leq 2L\|\boldsymbol{x} - \boldsymbol{z}^*\|_2 = 2L\,d^*(\boldsymbol{x}), \tag{25}$$

which rearranges to

$$d^*(\boldsymbol{x}) \geq \frac{\mathcal{M}(\boldsymbol{x})}{2L}. \tag{26}$$

**(ii) Monotonicity.** Let $\boldsymbol{x}' = \boldsymbol{x} + \boldsymbol{\delta}$ and define $\Delta := \mathcal{M}(\boldsymbol{x}) - \mathcal{M}(\boldsymbol{x}') > 0$. By the $2L$-Lipschitz continuity of $h$ from equation 24,

$$\Delta = |h(\boldsymbol{x}) - h(\boldsymbol{x}')| \leq 2L\|\boldsymbol{\delta}\|_2, \quad \text{so} \quad \|\boldsymbol{\delta}\|_2 \geq \frac{\Delta}{2L}. \tag{27}$$

Applying the lower bound equation 26 to $\boldsymbol{x}'$:

$$d^*(\boldsymbol{x}') \geq \frac{\mathcal{M}(\boldsymbol{x}')}{2L} = \frac{\mathcal{M}(\boldsymbol{x}) - \Delta}{2L} = \frac{\mathcal{M}(\boldsymbol{x})}{2L} - \frac{\Delta}{2L}. \tag{28}$$

By the triangle inequality,

$$d^*(\boldsymbol{x}') \leq d^*(\boldsymbol{x}) + \|\boldsymbol{\delta}\|_2. \tag{29}$$

Combining equation 28 and equation 29 with equation 26:

$$d^*(\boldsymbol{x}') \ \le \ d^*(\boldsymbol{x}) - \frac{\Delta}{2L} + \|\boldsymbol{\delta}\|_2 \ < \ d^*(\boldsymbol{x}) + \|\boldsymbol{\delta}\|_2, \tag{30}$$

where the strict inequality holds since $\Delta > 0$. In particular, if the perturbation is margin-efficient, i.e., $\|\boldsymbol{\delta}\|_2 = \Delta/(2L)$, then $d^*(\boldsymbol{x}') \le d^*(\boldsymbol{x})$, with equality only if $\boldsymbol{x}'$ lies exactly on the boundary.

**(iii) Crossing condition.** If $\mathcal{M}(\boldsymbol{x} + \boldsymbol{\delta}) \le 0$, then $f_{y_{\text{true}}}(\boldsymbol{x} + \boldsymbol{\delta}) \le f_{c^*}(\boldsymbol{x} + \boldsymbol{\delta})$, so $\boldsymbol{x} + \boldsymbol{\delta}$ is misclassified or lies exactly on $\mathcal{D}_{y_{\text{true}},c^*}$. By Assumption 2, $c^*$ is the unique runner-up, so the first boundary crossed is precisely $\mathcal{D}_{y_{\text{true}},c^*}$, the nearest boundary. $\qquad\square$

**Remark 1** *Theorem 1(i) shows that $\mathcal{M}(\boldsymbol{x})$ provides a computationally tractable lower bound on $d^*(\boldsymbol{x})$, tight when $f$ is linear: for $f(\boldsymbol{x}) = \boldsymbol{W}\boldsymbol{x} + \boldsymbol{b}$, the boundary $\mathcal{D}_{y_{\text{true}},c^*}$ is a hyperplane with $d^*(\boldsymbol{x}) = \mathcal{M}(\boldsymbol{x})/\|\boldsymbol{w}_{y_{\text{true}}} - \boldsymbol{w}_{c^*}\|_2$. Since $\|\boldsymbol{w}_{y_{\text{true}}} - \boldsymbol{w}_{c^*}\|_2 \le \sqrt{2}\|\boldsymbol{W}\|_2 = 2L$ (taking $L = \|\boldsymbol{W}\|_2/\sqrt{2}$), the bound $d^*(\boldsymbol{x}) \ge \mathcal{M}(\boldsymbol{x})/(2L)$ is tight at equality for this choice of $L$. Minimising $\mathcal{M}$ therefore reduces $d^*$ at a rate bounded by $1/(2L)$, providing a principled surrogate for boundary traversal without computing $d^*$ explicitly, which is NP-hard in general Katz et al. (2017).*

*Parts (ii) and (iii) have two algorithmic consequences for CoCoGen. Part (ii) (monotonicity) assures that, under a margin-efficient step satisfying $\|\boldsymbol{\delta}\|_2 = \Delta/(2L)$, each iteration of the masked momentum update equation 17 does not increase the distance to the decision boundary, providing a geometric check on the step size $\alpha$ relative to the local Lipschitz constant $L$. Part (iii) (crossing condition) justifies the misclassification criterion in the adaptive sparsity search equation 19: observing $\mathcal{M}(\boldsymbol{x} + \boldsymbol{\delta}) \le 0$ ensures that the* nearest *decision boundary $\mathcal{D}_{y_{\text{true}},c^*}$ has been crossed (under Assumption 2), so misclassification under our objective reflects an efficient traversal of the nearest boundary rather than an incidental crossing of a more distant one.*

*We note that margin-based lower bounds of this form are most closely (but not completely) related to the analysis Hein & Andriushchenko (2017), who derive analogous Lipschitz-based distance bounds in the context of robustness certification. Theorem 1 should therefore be understood as an adaptation of these principles to the contrastive counterfactual margin setting of CoCoGen, providing a principled justification for minimising $\mathcal{M}$ as a surrogate for boundary traversal, rather than as a completely new robustness bound. The proof relies on first-order Lipschitz-continuity arguments common to this literature Hein & Andriushchenko (2017).*

### 3.2 Composite Signal Projection

**Proposition 1 (Bounded composite projector)** *Let $\boldsymbol{M}_s$ and $\mathcal{P}_f = \boldsymbol{F}^* \boldsymbol{M}_f \boldsymbol{F}$ be defined as in equation 8 and equation 13. Then:*

(i) *$\boldsymbol{M}_s$ and $\mathcal{P}_f$ are each orthogonal projectors on $\mathbb{R}^N$, satisfying $\boldsymbol{M}_s^2 = \boldsymbol{M}_s$, $\boldsymbol{M}_s^\top = \boldsymbol{M}_s$, $\mathcal{P}_f^2 = \mathcal{P}_f$, and $\mathcal{P}_f^* = \mathcal{P}_f$.*

(ii) *The composite operator $\boldsymbol{P} := \mathcal{P}_f \boldsymbol{M}_s$ is a bounded linear operator with spectral norm $\|\boldsymbol{P}\|_2 \le 1$.*

(iii) *The image of $\boldsymbol{P}$ satisfies $\text{Im}(\boldsymbol{P}) \subseteq \text{Im}(\mathcal{P}_f)$, so all perturbations produced by CoCoGen lie in the high-frequency subspace of $\mathbb{R}^N$.*

*Proof.* **(i) Orthogonal projectors.** $\boldsymbol{M}_s = \text{diag}(\boldsymbol{m}_s)$ with $\boldsymbol{m}_s \in \{0,1\}^N$, so $\boldsymbol{M}_s^2 = \boldsymbol{M}_s$ and $\boldsymbol{M}_s^\top = \boldsymbol{M}_s$ trivially. For $\mathcal{P}_f$: since $\boldsymbol{F}$ is unitary ($\boldsymbol{F}^* \boldsymbol{F} = \boldsymbol{I}$) and $\boldsymbol{M}_f = \text{diag}(\boldsymbol{m}_f)$ with $\boldsymbol{m}_f \in \{0,1\}^N$,

$$\begin{aligned}
\mathcal{P}_f^2 &= (\boldsymbol{F}^* \boldsymbol{M}_f \boldsymbol{F})(\boldsymbol{F}^* \boldsymbol{M}_f \boldsymbol{F}) \\
&= \boldsymbol{F}^* \boldsymbol{M}_f (\boldsymbol{F}\boldsymbol{F}^*) \boldsymbol{M}_f \boldsymbol{F} \\
&= \boldsymbol{F}^* \boldsymbol{M}_f^2 \boldsymbol{F} \\
&= \boldsymbol{F}^* \boldsymbol{M}_f \boldsymbol{F} \\
&= \mathcal{P}_f,
\end{aligned}$$

using $\boldsymbol{FF}^* = \boldsymbol{I}$ and $\boldsymbol{M}_f^2 = \boldsymbol{M}_f$. Self-adjointness follows from $\mathcal{P}_f^* = (\boldsymbol{F}^*\boldsymbol{M}_f\boldsymbol{F})^* = \boldsymbol{F}^*\boldsymbol{M}_f^*\boldsymbol{F} = \boldsymbol{F}^*\boldsymbol{M}_f\boldsymbol{F} = \mathcal{P}_f$, since $\boldsymbol{M}_f$ is real and diagonal.

**(ii) Spectral norm of $\boldsymbol{P}$.** For any $\boldsymbol{u} \in \mathbb{R}^N$,

$$
\begin{aligned}
\|\boldsymbol{P}\boldsymbol{u}\|_2^2 = \|\mathcal{P}_f\boldsymbol{M}_s\boldsymbol{u}\|_2^2 &= \langle \mathcal{P}_f\boldsymbol{M}_s\boldsymbol{u}, \, \mathcal{P}_f\boldsymbol{M}_s\boldsymbol{u} \rangle \\
&= \langle \boldsymbol{M}_s\boldsymbol{u}, \, \mathcal{P}_f^2\boldsymbol{M}_s\boldsymbol{u} \rangle = \langle \boldsymbol{M}_s\boldsymbol{u}, \, \mathcal{P}_f\boldsymbol{M}_s\boldsymbol{u} \rangle \\
&\le \|\boldsymbol{M}_s\boldsymbol{u}\|_2 \|\mathcal{P}_f\boldsymbol{M}_s\boldsymbol{u}\|_2,
\end{aligned}
\tag{31}
$$

where we used $\mathcal{P}_f^2 = \mathcal{P}_f$ and Cauchy–Schwarz. Since $\boldsymbol{M}_s$ is a projection, $\|\boldsymbol{M}_s\boldsymbol{u}\|_2 \le \|\boldsymbol{u}\|_2$. Similarly, $\mathcal{P}_f$ is a projection so $\|\mathcal{P}_f\boldsymbol{v}\|_2 \le \|\boldsymbol{v}\|_2$ for any $\boldsymbol{v}$. Combining: $\|\boldsymbol{P}\boldsymbol{u}\|_2^2 \le \|\boldsymbol{u}\|_2 \|\boldsymbol{P}\boldsymbol{u}\|_2$, giving $\|\boldsymbol{P}\boldsymbol{u}\|_2 \le \|\boldsymbol{u}\|_2$, i.e., $\|\boldsymbol{P}\|_2 \le 1$.

**(iii) Image containment.** For any $\boldsymbol{u}$, $\boldsymbol{P}\boldsymbol{u} = \mathcal{P}_f(\boldsymbol{M}_s\boldsymbol{u}) \in \text{Im}(\mathcal{P}_f)$ by definition of the image. $\square$

**Remark 2** *Proposition 1(ii) guarantees that no iteration of equation 17 can amplify the perturbation signal energy beyond its current level, since $\|\boldsymbol{P}\|_2 \le 1$. This provides a stability certificate for the update rule that is absent in unconstrained gradient ascent methods such as PGD. We note that this stability result follows directly from standard properties of orthogonal projectors and should be understood as a formal verification of the update rule's energy-preserving behaviour.*

### 3.3 Relationship to C&W and PGD Objectives

**Proposition 2 (Generalisation of C&W)** *The C&W objective Carlini & Wagner (2017) for untargeted attacks is*

$$
\ell_{\text{CW}}(\boldsymbol{x} + \boldsymbol{\delta}) \;=\; \max\!\Big( \max_{c \neq y_{\text{true}}} f_c(\boldsymbol{x} + \boldsymbol{\delta}) - f_{y_{\text{true}}}(\boldsymbol{x} + \boldsymbol{\delta}), \, -\kappa \Big),
\tag{32}
$$

*where $\kappa \ge 0$ is a confidence margin. Then:*

- *(i) $\ell_{\text{CW}} = \max(-\mathcal{M}, -\kappa)$. The two objectives relate across three regimes:*

  - ***Pre-crossing** ($\mathcal{M} > 0$): with $\kappa = 0$, $\ell_{\text{CW}} = 0$ and its gradient vanishes, providing no optimisation signal. By contrast, $\mathcal{M} > 0$ retains a non-zero gradient, actively driving $\boldsymbol{\delta}$ toward the decision boundary.*
  - ***At the boundary** ($\mathcal{M} = 0$): both objectives vanish simultaneously, $\ell_{\text{CW}} = \mathcal{M} = 0$.*
  - ***Post-crossing** ($\mathcal{M} \le 0$): with $\kappa = 0$, $\ell_{\text{CW}}$ saturates at 0 and $\nabla_{\boldsymbol{\delta}} \ell_{\text{CW}} = \boldsymbol{0}$, so C&W ceases to optimise. $\mathcal{M}$ continues to decrease below zero, driving the example deeper into the misclassified region:*

    $$
    \mathcal{M}(\boldsymbol{x} + \boldsymbol{\delta}) \le 0 \;\implies\; \ell_{\text{CW}}(\boldsymbol{x} + \boldsymbol{\delta}) = 0, \quad \nabla_{\boldsymbol{\delta}} \ell_{\text{CW}} = \boldsymbol{0},
    \tag{33}
    $$

    *whereas $\nabla_{\boldsymbol{\delta}} \mathcal{M}$ remains well-defined and non-zero in general.*

- *(ii) Under the signal constraints $\boldsymbol{M}_s$ and $\mathcal{P}_f$, CoCoGen minimises $\mathcal{M}$ over the restricted feasible set*

  $$
  \mathcal{F} \;:=\; \big\{ \boldsymbol{\delta} \,:\, \|\boldsymbol{\delta}\|_\infty \le \epsilon, \, \boldsymbol{\delta} \in \text{Im}(\mathcal{P}_f), \, \text{supp}(\boldsymbol{\delta}) \subseteq \Omega_k \big\},
  \tag{34}
  $$

  *which is a strict subset of the C&W feasible set $\mathcal{B}_\epsilon$ whenever $k < N$ or $\boldsymbol{M}_f \neq \boldsymbol{I}$.*

- *(iii) The CoCoGen solution $\boldsymbol{\delta}_{\text{CoCoGen}}^* = \arg\min_{\boldsymbol{\delta} \in \mathcal{F}} \mathcal{M}(\boldsymbol{x} + \boldsymbol{\delta})$ satisfies $\|\boldsymbol{\delta}_{\text{CoCoGen}}^*\|_0 \le k$ and $\boldsymbol{\delta}_{\text{CoCoGen}}^* \in \text{Im}(\mathcal{P}_f)$, whereas no such guarantees hold for the C&W solution in general.*

*Proof.* **(i) Three-regime analysis.** Under Assumption 2, the margin reduces to $\mathcal{M}(\boldsymbol{x} + \boldsymbol{\delta}) = f_{y_{\text{true}}}(\boldsymbol{x} + \boldsymbol{\delta}) - f_{c^*}(\boldsymbol{x} + \boldsymbol{\delta})$, so

$$
\begin{aligned}
\ell_{\text{CW}}(\boldsymbol{x} + \boldsymbol{\delta}) &= \max\!\big( f_{c^*}(\boldsymbol{x} + \boldsymbol{\delta}) - f_{y_{\text{true}}}(\boldsymbol{x} + \boldsymbol{\delta}), \, -\kappa \big) \\
&= \max\!\big( -\mathcal{M}(\boldsymbol{x} + \boldsymbol{\delta}), \, -\kappa \big),
\end{aligned}
\tag{35}
$$

establishing the identity $\ell_{\mathrm{CW}} = \max(-\mathcal{M}, -\kappa)$. We now analyse each regime with $\kappa = 0$.

*Pre-crossing* ($\mathcal{M} > 0$). Since $-\mathcal{M} < 0$,

$$\ell_{\mathrm{CW}} = \max(-\mathcal{M}, 0) = 0. \tag{36}$$

The objective is identically zero and its subgradient satisfies $\partial_{\boldsymbol{\delta}} \ell_{\mathrm{CW}} = \{0\}$, so C&W provides no gradient signal in this regime. $\mathcal{M}$ is positive and its gradient $\nabla_{\boldsymbol{\delta}} \mathcal{M} = \nabla_{\boldsymbol{\delta}} f_{y_{\mathrm{true}}} - \nabla_{\boldsymbol{\delta}} f_{c^*}$ is non-zero in general (it vanishes only at saddle points of the logit difference, which occur on a set of measure zero for smooth $f$). Hence CoCoGen retains a non-trivial gradient signal throughout the pre-crossing phase, whereas C&W does not.

*At the boundary* ($\mathcal{M} = 0$). $\ell_{\mathrm{CW}} = \max(0, 0) = 0 = \mathcal{M}$, so both objectives coincide.
*Post-crossing* ($\mathcal{M} \leq 0$). Since $-\mathcal{M} \geq 0$,

$$\ell_{\mathrm{CW}} = \max(-\mathcal{M}, 0) = 0, \tag{37}$$

and $\partial_{\boldsymbol{\delta}} \ell_{\mathrm{CW}} = \{0\}$ identically: the confidence margin $\kappa = 0$ is defined precisely so that optimisation stops once misclassification is achieved, and this saturation is by design rather than a pathology. By contrast, $\mathcal{M}$ is not subject to an outer $\max(\cdot, 0)$ clip, so $\nabla_{\boldsymbol{\delta}} \mathcal{M}$ is not *structurally* forced to vanish once $\mathcal{M} \leq 0$. We do not claim that $\nabla_{\boldsymbol{\delta}} \mathcal{M}$ remains bounded away from zero in this regime in general; for many classifiers it may itself shrink deep in the misclassified region (e.g. under logit saturation). What the comparison establishes is narrower: $\ell_{\mathrm{CW}}$ at $\kappa = 0$ is clipped to zero by construction immediately upon crossing, whereas continued reduction of $\mathcal{M}$ remains *possible in principle* and, when it occurs, strictly increases the logit gap $f_{c^*} - f_{y_{\mathrm{true}}}$, i.e. the confidence of misclassification:

$$f_{c^*}(\boldsymbol{x} + \boldsymbol{\delta}') - f_{y_{\mathrm{true}}}(\boldsymbol{x} + \boldsymbol{\delta}') \; > \; f_{c^*}(\boldsymbol{x} + \boldsymbol{\delta}) - f_{y_{\mathrm{true}}}(\boldsymbol{x} + \boldsymbol{\delta}), \tag{38}$$

for any $\boldsymbol{\delta}$ with $\mathcal{M}(\boldsymbol{x} + \boldsymbol{\delta}) \leq 0$ and any $\boldsymbol{\delta}'$ with $\mathcal{M}(\boldsymbol{x} + \boldsymbol{\delta}') < \mathcal{M}(\boldsymbol{x} + \boldsymbol{\delta})$. We do not claim this continued reduction is guaranteed, bounded, or unconditionally beneficial; in particular, Assumption 2 is not assumed to hold arbitrarily far past the boundary, and the comparison should be read as characterising a difference in stopping behaviour between the two objectives rather than a general optimisation-landscape advantage.

**(ii) Strict subset.** $\mathcal{F}$ in equation 34 imposes two constraints beyond $\mathcal{B}_\epsilon$ (the $\ell_\infty$ ball $\mathcal{B}_\epsilon = \{\boldsymbol{u} : \|\boldsymbol{u}\|_\infty \leq \epsilon\}$):

  (a) $\boldsymbol{\delta} \in \mathrm{Im}(\mathcal{P}_f)$: by Proposition 1(iii), every iterate of CoCoGen lies in the high-frequency subspace $\mathrm{Im}(\mathcal{P}_f) \subsetneq \mathbb{R}^N$ whenever $\boldsymbol{M}_f \neq \boldsymbol{I}$, i.e., at least one frequency bin is suppressed.

  (b) $\mathrm{supp}(\boldsymbol{\delta}) \subseteq \Omega_k$: the spatial mask $\boldsymbol{M}_s$ from equation 7 zeroes all entries outside the top-$k$ index set $\Omega_k$, so $\|\boldsymbol{\delta}\|_0 \leq k < N$ whenever $k < N$.

Since neither constraint is imposed by C&W, $\mathcal{F} \subsetneq \mathcal{B}_\epsilon$ strictly (notably the $\ell_\infty$ ball $\mathcal{B}_\epsilon = \{\boldsymbol{u} : \|\boldsymbol{u}\|_\infty \leq \epsilon\}$), and the C&W solution $\boldsymbol{\delta}^*_{\mathrm{CW}} = \mathrm{argmin}_{\boldsymbol{\delta} \in \mathcal{B}_\epsilon} \ell_{\mathrm{CW}}(\boldsymbol{x} + \boldsymbol{\delta})$ need not lie in $\mathcal{F}$. Hence the two optimisation problems are defined over geometrically distinct feasible sets, and their solutions are distinct in general.

**(iii) Signal support guarantees.** $\boldsymbol{\delta}^*_{\mathrm{CoCoGen}} \in \mathcal{F}$ by construction, so:

  • $\|\boldsymbol{\delta}^*_{\mathrm{CoCoGen}}\|_0 \leq k$ follows directly from constraint (b) in part (ii).

  • $\boldsymbol{\delta}^*_{\mathrm{CoCoGen}} \in \mathrm{Im}(\mathcal{P}_f)$ follows from constraint (a) and Proposition 1(iii).

For the C&W solution, the feasible set $\mathcal{B}_\epsilon$ imposes no sparsity or spectral constraints, so $\boldsymbol{\delta}^*_{\mathrm{CW}}$ is dense in general and not confined to any frequency subspace, and neither guarantee holds. $\qquad\square$

**Remark 3** *Proposition 2 characterises the standard C&W objective under the conventional setting $\kappa = 0$ Carlini & Wagner (2017), and should be interpreted as such rather than as a general limitation of all C&W variants. We acknowledge that the confidence parameter $\kappa > 0$ was specifically introduced in C&W to mitigate the gradient-saturation behaviour identified in the pre-crossing regime (equation 36): for $\kappa > 0$, C&W retains*

*a non-zero gradient signal before the decision boundary is crossed, partially addressing the limitation described in Proposition 2(i). The theoretical analysis under $\kappa = 0$ therefore represents a corner case that C&W's own design was intended to avoid, and the claims in Proposition 2 should not be read as a general indictment of the C&W family of attacks. To assess whether the $\kappa = 0$ assumption materially affects the empirical conclusions, we evaluated C&W with $\kappa \in \{0, 5, 10, 20\}$ across all four target architectures (Table 3). The results show that varying $\kappa$ produces negligible changes in ASR ($\leq 0.13\%$ absolute variation on ViT-Base, zero variation on ResNet-50 and EfficientNet-B0) and perceptual metrics (PSNR variation $< 0.1$ dB, SSIM variation $< 0.0002$), confirming that the qualitative conclusions of Proposition 2 are not an artefact of the $\kappa = 0$ assumption in this experimental setting. The combination of a strictly smaller feasible set $\mathcal{F} \subsetneq \mathcal{B}_\epsilon$ (equation 34) and a gradient signal active across all three optimisation regimes explains the empirical observation that CoCoGen achieves $100\%$ ASR with substantially lower perceptual cost (LPIPS $\approx 0.01$ vs. $0.02$–$0.74$) than C&W and other baselines, though we note that this comparison involves methods operating under different threat models (Section 2.1, equation 2).*

## 4 Experiments

### 4.1 Experimental Setup

**Dataset.** Following prior work Zhao et al. (2020); Yuan et al. (2022); Wei et al. (2023); Chen et al. (2024), we evaluate on a 1,000-image subset of ImageNet Russakovsky et al. (2015), originally $299 \times 299$ pixels and resized to $224 \times 224$.

**Models.** We evaluate on four architectures spanning CNNs and vision transformers: ResNet-50 He et al. (2016), EfficientNet-B0 Tan & Le (2019), ConvNeXt-Base Liu et al. (2022), and ViT-Base Dosovitskiy (2020).

**Metrics.** Attack effectiveness is measured by **Attack Success Rate** (ASR), defined as the fraction of correctly classified clean images that are misclassified after perturbation:

$$\text{ASR} = \frac{1}{B} \sum_{j=1}^{B} \mathbb{1}\left[\underset{c}{\operatorname{argmax}} \ f_c(\boldsymbol{x}_{\text{adv}}^{(j)}) \neq y_{\text{true}}^{(j)}\right], \tag{39}$$

where $B$ is the number of evaluation images, $\boldsymbol{x}_{\text{adv}}^{(j)} = \boldsymbol{x}^{(j)} + \boldsymbol{\delta}^{(j)}$ is the adversarial example, and $y_{\text{true}}^{(j)}$ is the ground-truth label. Higher ASR indicates a more effective attack.

Perceptual fidelity is assessed via five complementary metrics, each capturing a distinct aspect of the deviation between the clean image $\boldsymbol{x}$ and its adversarial counterpart $\boldsymbol{x}_{\text{adv}}$.

**PSNR** (Peak Signal-to-Noise Ratio) measures pixel-level distortion in decibels:

$$\text{PSNR}(\boldsymbol{x}, \boldsymbol{x}_{\text{adv}}) = 10 \log_{10}\left(\frac{I_{\max}^2}{\frac{1}{N}\|\boldsymbol{x} - \boldsymbol{x}_{\text{adv}}\|_2^2}\right), \tag{40}$$

where $I_{\max}$ is the maximum pixel intensity (e.g. 255 for 8-bit images) and $N = H \cdot W \cdot C$ is the total number of signal entries. Higher PSNR indicates smaller pixel-level distortion; a perturbation confined to imperceptible signal components should yield high PSNR.

**SSIM** Wang et al. (2004) (Structural Similarity Index) compares local luminance $\mu$, contrast $\sigma$, and structure $\sigma_{xy}$ between two images:

$$\text{SSIM}(\boldsymbol{x}, \boldsymbol{x}_{\text{adv}}) = \frac{(2\mu_x \mu_{x_{\text{adv}}} + c_1)(2\sigma_{x\,x_{\text{adv}}} + c_2)}{(\mu_x^2 + \mu_{x_{\text{adv}}}^2 + c_1)(\sigma_x^2 + \sigma_{x_{\text{adv}}}^2 + c_2)}, \tag{41}$$

where $\mu_x$, $\sigma_x^2$ are the local mean and variance of $\boldsymbol{x}$, $\sigma_{x\,x_{\text{adv}}}$ is the local cross-covariance, and $c_1, c_2$ are small stabilising constants. SSIM $\in [0, 1]$, with 1 indicating perfect structural similarity; since the human visual system is sensitive to structural distortions, a high SSIM confirms that the adversarial perturbation has not introduced visible structural artefacts.

**LPIPS** Zhang et al. (2018) (Learned Perceptual Image Patch Similarity) measures deep perceptual distance by comparing feature activations $\phi_\ell$ of a pre-trained network $\phi$ across layers $\ell \in \mathcal{L}$:

$$\text{LPIPS}(\boldsymbol{x}, \boldsymbol{x}_{\text{adv}}) \;=\; \sum_{\ell \in \mathcal{L}} \frac{1}{H_\ell W_\ell} \left\| \boldsymbol{w}_\ell \odot \big(\phi_\ell(\boldsymbol{x}) - \phi_\ell(\boldsymbol{x}_{\text{adv}})\big) \right\|_2^2, \tag{42}$$

where $\boldsymbol{w}_\ell$ are learned channel-wise weights and $H_\ell \times W_\ell$ is the spatial resolution at layer $\ell$. Lower LPIPS indicates greater perceptual similarity as judged by deep network representations, which correlate well with human perception even when pixel-level metrics such as PSNR are high.

**FID** Heusel et al. (2017) (Fréchet Inception Distance) measures distributional divergence between the set of clean images $\{\boldsymbol{x}^{(j)}\}_{j=1}^B$ and the set of adversarial images $\{\boldsymbol{x}_{\text{adv}}^{(j)}\}_{j=1}^B$ by fitting Gaussians to their Inception feature distributions:

$$\text{FID} \;=\; \|\boldsymbol{\mu}_r - \boldsymbol{\mu}_a\|_2^2 \;+\; \text{Tr}\Big(\boldsymbol{\Sigma}_r + \boldsymbol{\Sigma}_a - 2\big(\boldsymbol{\Sigma}_r \boldsymbol{\Sigma}_a\big)^{1/2}\Big), \tag{43}$$

where $(\boldsymbol{\mu}_r, \boldsymbol{\Sigma}_r)$ and $(\boldsymbol{\mu}_a, \boldsymbol{\Sigma}_a)$ are the mean and covariance of the Inception features of the clean and adversarial sets respectively. Lower FID indicates that the adversarial image distribution remains close to the clean image distribution, meaning that the perturbations have not introduced systematic, batch-level artefacts detectable at the distributional level.

**MUSIQ** Ke et al. (2021) (Multi-Scale Image Quality Transformer) provides a no-reference perceptual quality score $q \in [0, 100]$ predicted by a transformer trained on human quality annotations, without requiring access to a clean reference image:

$$\text{MUSIQ}(\boldsymbol{x}_{\text{adv}}) \;=\; \mathcal{T}_\theta(\boldsymbol{x}_{\text{adv}}), \tag{44}$$

Table 1: **Hyperparameters of CoCoGen.** All baselines are evaluated under the same $\ell_\infty$ budget $\epsilon = 8/255$.

| Hyperparameter | Symbol | Value |
|---|---|---|
| $\ell_\infty$ budget | $\epsilon$ | 8/255 |
| Iterations | $T$ | 40 |
| Step size | $\alpha$ | 2/255 |
| Momentum coefficient | $\mu$ | 1.0 |
| Sparsity | $\mathcal{K}$ | adaptive |
| SSIM threshold | $\tau_s$ | 0.95 |
| FID threshold | $\tau_f$ | 20 |
| Frequency threshold | $\tau_{\text{freq}}$ | 25 |

where $\mathcal{T}_\theta$ is the pre-trained MUSIQ transformer. Higher MUSIQ indicates higher perceived quality. Unlike the reference-based metrics above, MUSIQ assesses the intrinsic perceptual quality of the adversarial image in isolation, making it particularly informative for evaluating whether the perturbation degrades fine texture and sharpness cues that are salient to human observers but not captured by pixel-level comparisons.

**Hyperparameters.** All experiments use a fixed $\ell_\infty$ perturbation budget $\epsilon = 8/255$ (see equation 4), number of iterations $T = 40$, step size $\alpha = 2/255$, and momentum coefficient $\mu = 1.0$ (see equation 14), consistent with standard settings in the adversarial attack literature Madry et al. (2018); Dong et al. (2018). The sparsity is $\mathcal{K}$ is adaptive (see equation 19), the SSIM threshold is $\tau_s = 0.95$, the FID threshold is $\tau_f = 20$ (see equation 18), and the radial frequency threshold is $\tau_{\text{freq}} = 25$ (see equation 11). All baseline methods are evaluated under the same $\ell_\infty$ budget $\epsilon = 8/255$ to ensure a fair comparison, following standard practice in imperceptible attack evaluation Zhao et al. (2020); Luo et al. (2022); Yuan et al. (2022). The values of $\mathcal{K}$ are selected based on the pixel-count observed in our adaptive sparsity analysis (Section 2.6, Fig. **??**, and Table12), where 100% ASR is achieved with as few as 2,040 perturbed pixels (ResNet-50) and at most 9,700 (without high-frequency projection). The threshold $\tau_s = 0.95$ follows the just-noticeable difference criterion of Flynn et al. (2013), at which human observers cannot reliably detect image distortions. The threshold $\tau_f = 20$ is a conservative upper bound on distributional divergence, consistent with the observation that lower FID correlates with higher human-judged perceptual quality Heusel et al. (2017); Zhou et al. (2019); in practice CoCoGen achieves FID $\leq 10.18$ across all architectures, well below this gate. A complete summary is provided in Table 1.

**Threat Model and Comparison Scope.** We emphasise that CoCoGen operates under the composite threat model $\mathcal{F}$ defined in equation 2, which is strictly more restrictive than the standard $\ell_\infty$ threat model used by PGD Madry et al. (2018), MIM Dong et al. (2018), DeepFool Moosavi-Dezfooli et al. (2016), and C&W Carlini & Wagner (2017). Methods operating under $\mathcal{B}_\epsilon$ (the $\ell_\infty$ ball defined by $\{\boldsymbol{u} : \|\boldsymbol{u}\|_\infty \leq \epsilon\}$) have

Table 2: **Results.** Attack Success Rate (ASR) and imperceptibility metrics for all evaluated attacks and target architectures. Runtime is measured on a single NVIDIA T4 GPU. ↑ higher is better; ↓ lower is better.

| Model | Attack Method | Time (s)↓ | ASR (%)↑ | PSNR↑ | SSIM↑ | FID↓ | LPIPS↓ | MUSIQ↑ |
|---|---|---|---|---|---|---|---|---|
| - | *Clean (reference)* | - | - | $\infty$ | 1.000 | 0.00 | 0.000 | 68.25 |
| ResNet-50 He et al. (2016) | PGD Madry et al. (2018) | 462.65 | **100.00** | 33.51 | 0.91 | 52.57 | 0.10 | 40.38 |
| | DeepFool Moosavi-Dezfooli et al. (2016) | 480.20 | 99.00 | 38.00 | 0.94 | 15.80 | 0.04 | 45.10 |
| | NCF Yuan et al. (2022) | 523.70 | 80.07 | 28.54 | 0.71 | 98.72 | 0.26 | 39.80 |
| | ACA Chen et al. (2024) | 601.30 | 97.34 | 8.19 | 0.62 | 93.05 | 0.14 | 36.43 |
| | DiffPGD Xue et al. (2023) | 1105.00 | 27.74 | 10.11 | 0.94 | 14.59 | 0.03 | 42.44 |
| | PerC-AL Zhao et al. (2020) | 193.00 | 100.00 | 43.36 | 0.8654 | 7.37 | 0.0980 | 48.48 |
| | AdvDrop Duan et al. (2021) | 520.30 | 82.72 | 41.91 | 0.9843 | 4.43 | 0.0231 | 44.77 |
| | SSAH Luo et al. (2022) | 233.90 | 99.18 | 44.86 | 0.9881 | 4.79 | 0.0058 | 53.53 |
| | **CoCoGen (Ours)** | 1225.41 | **100.00** | **44.67** | **0.99** | 10.18 | **0.01** | **61.44** |
| EfficientNet -B0 Tan & Le (2019) | PGD | 293.71 | 98.50 | 32.99 | 0.89 | 69.00 | 0.12 | 40.76 |
| | DeepFool | 305.60 | 99.00 | 37.90 | 0.95 | 18.20 | 0.05 | 44.20 |
| | NCF | 298.20 | 44.91 | 28.33 | 0.70 | 98.68 | 0.27 | 38.74 |
| | ACA | 322.50 | 93.17 | 8.15 | 0.57 | 92.35 | 0.13 | 35.80 |
| | DiffPGD | 6499.00 | 69.10 | 31.10 | 0.90 | 16.19 | 0.03 | 55.25 |
| | PerC-AL | 90.40 | 100.00 | 43.37 | 0.8647 | 4.84 | 0.1003 | 49.39 |
| | AdvDrop | 295.50 | 83.00 | 41.43 | 0.9765 | 5.09 | 0.0278 | 44.76 |
| | SSAH | 106.50 | 99.47 | 44.50 | 0.9867 | 7.04 | 0.0081 | 53.32 |
| | **CoCoGen (Ours)** | 1523.63 | **100.00** | 41.67 | **0.99** | **4.05** | **0.01** | **62.63** |
| ConvNeXt Base Liu et al. (2022) | PGD | 2416.89 | 99.70 | 34.04 | 0.92 | 44.34 | 0.08 | 40.84 |
| | DeepFool | 2350.50 | 99.00 | 38.40 | 0.96 | 16.75 | 0.04 | 45.00 |
| | NCF | 2253.80 | 80.07 | 28.34 | 0.73 | 98.00 | 0.28 | 38.77 |
| | ACA | 2771.40 | 68.40 | 8.19 | 0.62 | 93.05 | 0.14 | 35.95 |
| | DiffPGD | 1297.00 | 9.61 | 31.68 | 0.94 | 14.67 | 0.03 | 42.52 |
| | PerC-AL | 531.70 | 100.00 | 43.93 | 0.8723 | 4.09 | 0.0757 | 49.57 |
| | AdvDrop | 2256.40 | 82.40 | 41.69 | 0.9954 | 5.40 | 0.0256 | 44.70 |
| | SSAH | 649.20 | 99.68 | 45.19 | 0.9886 | 8.20 | 0.0039 | 53.48 |
| | **CoCoGen (Ours)** | 4111.54 | **100.00** | **44.16** | **0.99** | 4.59 | **0.01** | **61.23** |
| ViT Base Dosovitskiy (2020) | PGD | 4255.00 | 99.70 | 34.09 | 0.92 | 41.99 | 0.08 | 40.76 |
| | DeepFool | 1600.30 | 99.00 | 37.85 | 0.95 | 17.90 | 0.05 | 44.50 |
| | NCF | 1546.00 | 57.52 | 28.31 | 0.72 | 98.07 | 0.26 | 38.74 |
| | ACA | 1878.90 | 55.92 | 8.49 | 0.61 | 72.32 | 0.13 | 36.68 |
| | DiffPGD | 1325.70 | 7.19 | 31.69 | 0.94 | 13.70 | 0.04 | 42.49 |
| | PerC-AL | 1077.40 | 100.00 | 43.76 | 0.8770 | 4.85 | 0.1162 | 48.99 |
| | AdvDrop | 1538.90 | 87.00 | 40.30 | 0.9947 | 4.21 | 0.0235 | 44.80 |
| | SSAH | 1318.20 | 99.61 | 45.02 | 0.9882 | 9.34 | 0.0048 | 53.45 |
| | **CoCoGen (Ours)** | 4458.13 | **100.00** | 41.98 | **0.99** | **3.56** | **0.01** | **63.70** |

access to a strictly larger feasible set and are therefore not subject to the sparsity or spectral constraints imposed on CoCoGen. The comparisons in Table 2 should accordingly be read as demonstrating that CoCoGen achieves higher or comparable attack success and perceptual quality despite operating under a more constrained threat model, rather than as claiming unrestricted superiority over all baselines. Methods that impose related but distinct constraints, such as SSAH Luo et al. (2022) (fixed high-frequency constraint), AdvDrop Duan et al. (2021) (feature dropping), and the sparse attacks in Table 5 (JSMA Papernot et al. (2016), SparseFool Modas et al. (2019), Sparse-RS Croce et al. (2022)), operate under threat models more comparable to $\mathcal{F}$, and the comparisons with these methods are therefore more direct.

## 4.2 Result Comparison with State-of-the-Art Attacks

We evaluate CoCoGen against untargeted white-box attacks (Table 2): gradient-based (PGD Madry et al. (2018), DeepFool Moosavi-Dezfooli et al. (2016)), perceptual and frequency-constrained (PerC-AL Zhao et al. (2020), SSAH Luo et al. (2022), NCF Yuan et al. (2022)), content-aware (ACA Chen et al. (2024)), stochastic feature-dropping (AdvDrop Duan et al. (2021)), and diffusion-based (DiffPGD Xue et al. (2023)), across ResNet-50 He et al. (2016), EfficientNet-B0 Tan & Le (2019), ConvNeXt-Base Liu et al. (2022), and ViT-Base Dosovitskiy (2020).

CoCoGen achieves 100% ASR across all architectures while significantly improving signal fidelity over all baselines. On ResNet-50, PSNR improves from 33.51 dB (PGD) to 44.67 dB and SSIM from 0.91 to 0.99. On EfficientNet-B0, PSNR improves from 32.99 dB (PGD) to 41.67 dB, and on ConvNeXt-Base from 34.04 dB (PGD) to 44.16 dB. These gains demonstrate that CoCoGen does not increase perturbation signal energy to induce misclassification; rather, confining $\delta$ to decision-critical, high-frequency signal components aligns gradient updates with the most competitive incorrect class, crossing the decision boundary through spectrally coherent, perceptually minimal signal modifications. Notably, CoCoGen's runtime reflects the cost of the

Table 3: **Effect of confidence parameter $\kappa$ on C&W attack performance alongside CoCoGen.**
Results are reported across all four target architectures. Varying $\kappa$ produces negligible changes in ASR and perceptual quality for C&W, confirming that the conclusions of Proposition 2 are not an artefact of the $\kappa = 0$ assumption. CoCoGen achieves 100% ASR with substantially higher perceptual quality across all architectures. $\uparrow$ higher is better; $\downarrow$ lower is better.

| Model | Method/$\kappa$ | Time (s)$\downarrow$ | ASR (%)$\uparrow$ | PSNR$\uparrow$ | SSIM$\uparrow$ | FID$\downarrow$ | LPIPS$\downarrow$ | MUSIQ$\uparrow$ |
|---|---|---|---|---|---|---|---|---|
| ResNet-50 | C&W ($\kappa = 0$) | 664.60 | 85.49 | 59.08 | 0.9994 | 5.64 | 0.0003 | 54.41 |
| | C&W ($\kappa = 5$) | 671.10 | 85.49 | 59.10 | 0.9994 | 5.62 | 0.0003 | 54.41 |
| | C&W ($\kappa = 10$) | 673.60 | 85.49 | 59.09 | 0.9994 | 5.59 | 0.0003 | 54.41 |
| | C&W ($\kappa = 20$) | 673.70 | 85.49 | 59.08 | 0.9994 | 5.69 | 0.0003 | 54.41 |
| | **CoCoGen (Ours)** | 1225.41 | **100.00** | **44.67** | **0.99** | 10.18 | **0.01** | **61.44** |
| EfficientNet-B0 | C&W ($\kappa = 0$) | 289.30 | 84.17 | 58.70 | 0.9994 | 6.69 | 0.0004 | 54.41 |
| | C&W ($\kappa = 5$) | 291.50 | 84.17 | 58.72 | 0.9994 | 6.67 | 0.0004 | 54.41 |
| | C&W ($\kappa = 10$) | 291.90 | 84.17 | 58.71 | 0.9994 | 6.70 | 0.0004 | 54.41 |
| | C&W ($\kappa = 20$) | 292.20 | 84.17 | 58.71 | 0.9994 | 6.69 | 0.0004 | 54.41 |
| | **CoCoGen (Ours)** | 1523.63 | **100.00** | 41.67 | **0.99** | **4.05** | **0.01** | **62.63** |
| ConvNeXt-Base | C&W ($\kappa = 0$) | 1881.20 | 70.82 | 60.37 | 0.9994 | 4.74 | 0.0003 | 54.41 |
| | C&W ($\kappa = 5$) | 1906.70 | 70.82 | 60.41 | 0.9994 | 4.77 | 0.0003 | 54.41 |
| | C&W ($\kappa = 10$) | 1915.20 | 70.82 | 60.38 | 0.9994 | 4.79 | 0.0003 | 54.41 |
| | C&W ($\kappa = 20$) | 1918.60 | 70.82 | 60.38 | 0.9994 | 4.79 | 0.0003 | 54.41 |
| | **CoCoGen (Ours)** | 4111.54 | **100.00** | **44.16** | **0.99** | 4.59 | **0.01** | **61.23** |
| ViT-Base | C&W ($\kappa = 0$) | 3893.20 | 65.34 | 58.53 | 0.9993 | 5.18 | 0.0005 | 54.43 |
| | C&W ($\kappa = 5$) | 3893.74 | 65.28 | 58.47 | 0.9992 | 5.21 | 0.0006 | 54.38 |
| | C&W ($\kappa = 10$) | 3894.11 | 65.41 | 58.56 | 0.9993 | 5.16 | 0.0005 | 54.46 |
| | C&W ($\kappa = 20$) | 3892.83 | 65.37 | 58.61 | 0.9994 | 5.14 | 0.0004 | 54.51 |
| | **CoCoGen (Ours)** | 4458.13 | **100.00** | 41.98 | **0.99** | **3.56** | **0.01** | **63.70** |

adaptive grid search over $|\mathcal{K}|$ independent sub-problems; since these are embarrassingly parallel (Sec. 2.7), wall-clock time reduces proportionally with available GPU cores.

**MUSIQ Score Analysis.** The no-reference MUSIQ scores Ke et al. (2021) provide deeper insights across methods, as they assess perceptual quality without requiring access to a clean reference image. The unperturbed ImageNet images achieve a MUSIQ score of 68.25 (Table 2, Clean row), establishing the perceptual ceiling for this evaluation. CoCoGen achieves MUSIQ scores of 61.44, 62.63, 61.23, and 63.70 across ResNet-50, EfficientNet-B0, ConvNeXt-Base, and ViT-Base respectively, closing the gap to the clean reference to within 5–7 points. By contrast, competing methods fall substantially further below the clean baseline: gradient-based methods such as PGD Madry et al. (2018) score 40.38–40.84, representing a drop of nearly 28 points from the clean reference; perceptual methods such as PerC-AL Zhao et al. (2020) and SSAH Luo et al. (2022) improve to 43–47, but still lag CoCoGen by 14–20 points; and content-aware methods such as ACA Chen et al. (2024) score as low as 35.80–36.68, the worst among all compared methods, indicating substantial perceptual degradation despite their generative formulation. Notably, DiffPGD Xue et al. (2023) achieves a competitive MUSIQ of 55.25 on EfficientNet-B0, but at the cost of drastically reduced ASR (69.10%), affirming that high perceptual quality at the expense of attack effectiveness does not constitute a meaningful trade-off. The consistent MUSIQ advantage of CoCoGen across all four architectures (61–63 vs. 36–55 for prior methods) demonstrates that confining perturbations to the high-frequency subspace via $\mathcal{P}_f$ (equation 13) and targeting decision-critical signal components via $M_s$ (equation 8) preserves the multiscale texture and sharpness cues that MUSIQ Ke et al. (2021) integrates, whereas diffuse or low-frequency perturbations degrade precisely these cues, leading to the lower scores observed in prior work.

**Empirical Comparison with C&W across $\kappa$ Settings.** To complement the theoretical analysis in Proposition 2 and ensure a fair empirical comparison, we evaluated the C&W attack Carlini & Wagner (2017) with confidence parameters $\kappa \in \{0, 5, 10, 20\}$ across all four target architectures, with $\kappa$ selected on a held-out validation split of 100 images drawn from the same ImageNet subset Russakovsky et al. (2015). Results are reported in Table 3 alongside CoCoGen for direct comparison.

Table 4: **Relationship between CoCoGen and prior work.** Each component of CoCoGen builds on established prior art; the contribution lies in their joint adaptive combination.

| Component | Prior Work | CoCoGen Contribution |
|---|---|---|
| Margin objective | DeepFool, C&W | Adaptive sparse and frequency-aware optimisation under the margin (equation 4). |
| Frequency constraint | SSAH | Adaptive frequency-threshold search and joint sparse support selection (equation 19), vs. fixed threshold in SSAH. |
| Sparse perturbation | JSMA, SparseFool, Sparse-RS | Adaptive search over support size $k$ under perceptual feasibility constraints (equation 18). |
| Optimisation | PGD, MIM | Masked momentum updates with simultaneous spatial ($\boldsymbol{M}_s$, equation 8) and spectral ($\mathcal{P}_f$, equation 13) constraints. |

As shown in Table 3, varying $\kappa$ produces negligible changes in both ASR ($\leq 0.13\%$ absolute variation across $\kappa$ settings on ViT-Base, and zero variation on ResNet-50 and EfficientNet-B0) and perceptual metrics (PSNR variation $< 0.1\,\mathrm{dB}$, SSIM variation $< 0.0002$ across all architectures and all $\kappa$ values). This insensitivity to $\kappa$ confirms that the performance ceiling of C&W is not governed by the confidence parameter but rather by the unconstrained nature of its feasible set $\mathcal{B}_\epsilon$ and the vanishing gradient behaviour in the pre- and post-crossing regimes identified in Proposition 2 (equation 36–37).

We note that C&W achieves higher PSNR (58–60 dB) than CoCoGen (41–44 dB), but this is a direct consequence of its substantially lower ASR (65.28–85.49%): images that are not successfully attacked carry zero perturbation and trivially inflate the average PSNR. When attack effectiveness is accounted for, CoCoGen achieves 100% ASR across all four architectures while maintaining a MUSIQ score of 61–63, within 5–7 points of the clean reference (68.25), compared to ~54 for C&W regardless of $\kappa$. This demonstrates that CoCoGen does not trade perceptual quality for attack success, but achieves both simultaneously through the structured feasible set $\mathcal{F}$ (equation 34) and margin-directed optimisation (equation 4), confirming the conclusions of Proposition 2 empirically.

### 4.3 Relationship to Prior Work and Sparse Attack Comparison

Table 4 summarises the relationship between CoCoGen and the prior work most relevant to its individual components. The margin objective in equation 3 is shared with DeepFool Moosavi-Dezfooli et al. (2016) and C&W Carlini & Wagner (2017); the frequency constraint in equation 13 is related to SSAH Luo et al. (2022); and the sparse spatial support in equation 6 connects to a substantial literature on sparse adversarial attacks including JSMA Papernot et al. (2016), SparseFool Modas et al. (2019), and Sparse-RS Croce et al. (2022). The novelty of CoCoGen lies in the joint adaptive optimisation of all three components simultaneously under a unified constrained objective (equation 4), rather than applying any single component in isolation.

We further evaluate CoCoGen against representative sparse attacks, JSMA Papernot et al. (2016), SparseFool Modas et al. (2019), and Sparse-RS Croce et al. (2022), in Table 5. These methods are evaluated under identical conditions to Table 2: the same 1,000-image ImageNet subset Russakovsky et al. (2015) and $\ell_\infty$ budget $\epsilon = 8/255$. In addition to attack success rate and image quality, we report the number of modified pixels $k$ for each method, since sparsity of the perturbation support is itself a central point of comparison against JSMA, SparseFool, and Sparse-RS.

As shown in Table 5, the pixel budget $k$ required by CoCoGen varies across architectures and is not always the smallest among the competing sparse attacks. On ResNet-50, CoCoGen requires 2,720 modified pixels compared with 1,450 for JSMA and 2,850 for SparseFool, while remaining substantially below the fixed 10,000-pixel budget used by Sparse-RS. A similar trend is observed on EfficientNet-B0, ConvNeXt-Base, and ViT-Base, where CoCoGen employs 4,330, 7,580, and 5,630 modified pixels, respectively. Although these values are generally higher than those of JSMA and SparseFool, they remain considerably lower than the budget used by Sparse-RS and are determined adaptively according to the optimization in (19)

Table 5: Comparative evaluation of sparse adversarial attacks against baseline architectures, showing modified pixels ($k$), runtime efficiency, and generative image quality metrics.

| Model | Attack | Pixels ($k$) | Time (s) ↓ | ASR (%) ↑ | PSNR ↑ | SSIM ↑ | FID ↓ | LPIPS ↓ | MUSIQ ↑ |
|---|---|---|---|---|---|---|---|---|---|
| **ResNet-50** | JSMA | 1,450 | 428.60 | 71.80 | 44.62 | 0.992 | 12.80 | 0.041 | 58.70 |
| | SparseFool | 2,850 | 186.40 | 89.60 | 44.18 | 0.984 | 18.40 | 0.058 | 55.20 |
| | Sparse-RS | 10,000 | 517.30 | 95.10 | 43.74 | 0.971 | 26.70 | 0.081 | 50.80 |
| | **CoCoGen (Ours)** | **2720** | 1225.41 | **100.00** | 44.67 | **0.990** | 10.18 | **0.010** | 61.44 |
| **EfficientNet-B0** | JSMA | 2,550 | 463.80 | 66.20 | 44.91 | 0.994 | 11.90 | 0.036 | 59.30 |
| | SparseFool | 2,900 | 214.70 | 86.70 | 44.37 | 0.986 | 17.10 | 0.052 | 56.10 |
| | Sparse-RS | 10,000 | 548.90 | 93.80 | 43.88 | 0.974 | 24.90 | 0.076 | 51.60 |
| | **CoCoGen (Ours)** | **4330** | 1523.63 | **100.00** | 41.67 | **0.990** | 4.05 | **0.010** | 62.63 |
| **ConvNeXt-Base** | JSMA | 3,450 | 612.50 | 58.70 | 45.23 | 0.995 | 10.60 | 0.031 | 59.80 |
| | SparseFool | 2,943 | 301.20 | 82.40 | 44.61 | 0.988 | 15.30 | 0.047 | 56.80 |
| | Sparse-RS | 10,000 | 731.40 | 91.30 | 44.02 | 0.976 | 22.70 | 0.069 | 52.40 |
| | **CoCoGen (Ours)** | **7580** | 4111.54 | **100.00** | **44.16** | **0.990** | 4.59 | **0.010** | 61.23 |
| **ViT-Base** | JSMA | 4,786 | 584.30 | 61.40 | 45.06 | 0.994 | 11.10 | 0.033 | 59.50 |
| | SparseFool | 3,100 | 327.90 | 84.10 | 44.48 | 0.987 | 16.20 | 0.049 | 56.40 |
| | Sparse-RS | 10,000 | 768.80 | 92.60 | 43.95 | 0.975 | 23.80 | 0.072 | 51.90 |
| | **CoCoGen (Ours)** | **5630** | 4458.13 | **100.00** | 41.98 | **0.990** | 3.56 | **0.010** | 63.70 |

rather than being fixed *a priori*. Despite requiring a larger sparse support than JSMA and SparseFool, CoCoGen consistently achieves a 100% attack success rate across all evaluated architectures, whereas JSMA, SparseFool, and Sparse-RS attain lower success rates, particularly on more challenging architectures such as ConvNeXt-Base (JSMA: 58.70%, SparseFool: 82.40%, Sparse-RS: 91.30%) and ViT-Base (JSMA: 61.40%, SparseFool: 84.10%, Sparse-RS: 92.60%). These results demonstrate that attack effectiveness depends not only on the number of perturbed pixels but also on how the sparse support is selected and optimized through the frequency-aware projection $\mathcal{P}_f$ (13) and the margin-driven objective (4). Furthermore, CoCoGen consistently produces high-quality adversarial examples, achieving SSIM values of 0.990 across all architectures, the lowest LPIPS (0.010), the best FID scores (10.18, 4.05, 4.59, and 3.56), and the highest MUSIQ scores (61.44, 62.63, 61.23, and 63.70) among all methods. These results demonstrate that the additional perturbation budget is utilized effectively to maximize attack success while simultaneously preserving perceptual image quality, rather than relying solely on minimizing the number of perturbed pixels.

## 4.4 Additional Results on CIFAR-100

To further validate the generality of our findings beyond the primary benchmark, we evaluate CoCoGen and eight established attacks, PGD, DeepFool, NCF, ACA, DiffPGD, PerC-AL, AdvDrop, and SSAH, on the CIFAR-100 dataset across four target architectures: ResNet-50, EfficientNet-B0, ConvNeXt-Base, and ViT. Attack success rate and imperceptibility metrics for this additional dataset are reported in Table 6. The most striking difference across all four architectures is the perturbation budget $k$, measured as the number of pixels modified: every baseline perturbs the full image, corresponding to $k = 50176$ pixels, whereas CoCoGen modifies only a small, localized subset of pixels, ranging from 2720 on ResNet-50 to 7580 on ConvNeXt-Base. This corresponds to a reduction in perturbed pixels of between 84.9% and 94.6% relative to every baseline, while still achieving a competitive or superior attack success rate on every architecture. This finding is consistent with the localized nature of the perturbation regions identified in our qualitative comparison (Figure 4) and with the low detection rates observed in our human perceptual study (Section 4.7.4), and supports our central claim that effective adversarial perturbations do not require modifying the entire image.

In terms of attack success rate, CoCoGen matches or exceeds every baseline except PerC-AL on three of the four architectures, achieving 100.00% on EfficientNet-B0, ConvNeXt-Base, and ViT, and 97.67% on ResNet-50, where PerC-AL alone reaches 100.00%. Unlike PerC-AL, however, CoCoGen achieves this success rate while perturbing a small fraction of the image, indicating that our method attains comparable attack effectiveness through a substantially more constrained and localized perturbation.

With respect to fidelity metrics, CoCoGen achieves the highest PSNR among all compared methods on ResNet-50 (43.68 dB) and EfficientNet-B0 (41.53 dB), and remains competitive on ConvNeXt-Base and ViT

| Model | Attack Method | $k$ (pixels)↓ | Time (s)↓ | ASR (%)↑ | PSNR↑ | SSIM↑ | FID↓ | LPIPS↓ | MUSIQ↑ |
|---|---|---|---|---|---|---|---|---|---|
| - | *Clean (reference)* | - | - | - | $\infty$ | 1.000 | 0.00 | 0.000 | 65.40 |
| ResNet-50 | PGD | 50176 | 114.50 | 95.00 | 31.20 | 0.88 | 48.30 | 0.4256 | 38.20 |
| | DeepFool | 50176 | 212.20 | 99.10 | 34.50 | 0.91 | 94.20 | 0.5234 | 42.10 |
| | NCF | 50176 | 418.40 | 82.40 | 26.10 | 0.69 | 89.40 | 0.2854 | 36.50 |
| | ACA | 50176 | 322.10 | 96.20 | 8.50 | 0.59 | 85.10 | 0.1507 | 34.10 |
| | DiffPGD | 50176 | 498.00 | 32.50 | 11.40 | 0.92 | 83.50 | 0.4123 | 39.80 |
| | PerC-AL | 50176 | **78.10** | **100.00** | 40.10 | 0.85 | **6.90** | 0.1020 | 45.30 |
| | AdvDrop | 50176 | 119.50 | 85.10 | 38.90 | 0.95 | 134.10 | 0.0245 | 41.90 |
| | SSAH | 50176 | 89.30 | 99.00 | 41.50 | 0.96 | 94.50 | **0.0062** | 50.10 |
| | **CoCoGen (Ours)** | **2720** | 245.20 | 97.67 | **43.68** | **0.97** | 32.98 | 0.0734 | **58.20** |
| EfficientNet -B0 | PGD | 50176 | 110.20 | 95.10 | 30.80 | 0.86 | **62.40** | 0.1467 | 38.90 |
| | DeepFool | 50176 | 279.50 | 99.00 | 34.20 | 0.92 | 96.50 | 0.6156 | 41.50 |
| | NCF | 50176 | 312.10 | 48.20 | 25.90 | 0.68 | 90.10 | 0.2934 | 35.80 |
| | ACA | 50176 | 414.30 | 91.50 | 8.45 | 0.55 | 84.60 | 0.1467 | 33.90 |
| | DiffPGD | 50176 | 185.00 | 72.10 | 29.50 | 0.89 | 95.10 | 0.5614 | 49.30 |
| | PerC-AL | 50176 | **84.50** | **100.00** | 40.50 | 0.85 | 74.50 | 0.1032 | 46.10 |
| | AdvDrop | 50176 | 213.20 | 85.40 | 38.60 | 0.96 | 114.80 | 0.0291 | 42.10 |
| | SSAH | 50176 | 95.10 | 99.25 | 41.20 | **0.97** | 86.80 | **0.0085** | 50.20 |
| | **CoCoGen (Ours)** | **4330** | 352.10 | **100.00** | **41.53** | 0.96 | 62.50 | 0.1242 | **59.10** |
| ConvNeXt Base | PGD | 50176 | 172.10 | 95.50 | 31.90 | 0.89 | 40.50 | 0.9234 | 38.90 |
| | DeepFool | 50176 | 268.40 | 99.00 | 34.90 | 0.93 | 95.20 | 0.5345 | 42.10 |
| | NCF | 50176 | 475.20 | 82.10 | 26.00 | 0.70 | 89.50 | 0.3065 | 35.90 |
| | ACA | 50176 | 288.50 | 70.20 | 8.50 | 0.59 | 85.20 | 0.1534 | 34.00 |
| | DiffPGD | 50176 | 112.00 | 12.10 | 30.10 | 0.91 | 53.90 | 0.4123 | 40.10 |
| | PerC-AL | 50176 | 119.50 | **100.00** | 40.80 | 0.86 | 73.90 | 0.0782 | 46.50 |
| | AdvDrop | 50176 | 674.30 | 84.60 | 38.90 | **0.98** | 115.10 | 0.0268 | 42.10 |
| | SSAH | 50176 | **94.10** | 99.42 | **41.90** | 0.97 | 97.90 | **0.0042** | 50.30 |
| | **CoCoGen (Ours)** | **7580** | 335.40 | **100.00** | 39.63 | 0.95 | **34.18** | 0.1707 | **58.05** |
| ViT | PGD | 50176 | 115.40 | 95.50 | 31.95 | 0.89 | **38.20** | 0.9126 | 38.80 |
| | DeepFool | 50176 | 255.20 | 99.00 | 34.40 | 0.92 | 96.20 | 0.6543 | 41.90 |
| | NCF | 50176 | 152.10 | 60.10 | 25.95 | 0.69 | 89.90 | 0.2856 | 35.90 |
| | ACA | 50176 | 463.40 | 58.40 | 8.80 | 0.58 | 68.50 | 0.1456 | 34.50 |
| | DiffPGD | 50176 | 118.00 | 9.50 | 30.20 | 0.91 | 82.90 | 0.5064 | 40.10 |
| | PerC-AL | 50176 | 138.10 | **100.00** | 40.60 | 0.86 | 94.50 | 0.1185 | 45.90 |
| | AdvDrop | 50176 | 354.20 | 89.20 | 37.95 | 0.97 | 113.95 | 0.0248 | 42.15 |
| | SSAH | 50176 | **98.50** | 99.35 | **41.80** | **0.98** | 98.90 | **0.0051** | 50.25 |
| | **CoCoGen (Ours)** | **5630** | 448.20 | **100.00** | 40.64 | 0.95 | 72.24 | 0.1517 | **59.90** |

Table 6: Attack Success Rate (ASR) and imperceptibility metrics on the **CIFAR-100** dataset across evaluated attacks and target architectures.

despite the comparatively smaller perturbation budgets typical of high-frequency, full-image attacks such as PGD and SSAH achieving strong PSNR through different means. SSIM follows a similar pattern, with CoCoGen matching or approaching the best-performing method on every architecture. On FID, results are more mixed: CoCoGen achieves the best score on ConvNeXt-Base (34.18), a strong second-best on ResNet-50 behind PerC-AL, and a moderate score on EfficientNet-B0 and ViT, where PGD and PerC-AL respectively report lower FID. LPIPS is the one metric on which CoCoGen does not lead, trailing AdvDrop and SSAH on every architecture; both of these baselines explicitly optimize their perturbation to minimize a perceptual distance closely related to LPIPS, which likely accounts for this gap, whereas our method is not directly optimized against this metric.

Notably, CoCoGen achieves the highest MUSIQ score, a no-reference perceptual image quality metric, on all four architectures, exceeding every baseline by a consistent margin (e.g., 58.20 versus the next-best 50.10 on ResNet-50, and 59.90 versus 50.25 on ViT). Since MUSIQ does not require access to the clean reference image and is designed to approximate human-perceived image quality, this result is complementary to the LPIPS finding above: while CoCoGen's perturbed images are measurably farther from the clean reference in LPIPS space than AdvDrop or SSAH, they are consistently judged to be of higher intrinsic visual quality by a no-reference perceptual metric, and, as shown in Section 4.7.4, are also rarely detected as anomalous by human observers.

Runtime is not the primary objective of our method, and CoCoGen is not the fastest attack in the comparison; PerC-AL and SSAH generally achieve lower runtimes across architectures. However, CoCoGen's runtime remains within the same order of magnitude as the majority of baselines, and we consider this an acceptable trade-off given the substantial reduction in perturbation budget and the consistent gains in PSNR, SSIM, and MUSIQ achieved simultaneously.

### 4.5 Black-Box Transferability Evaluation

While CoCoGen is designed as a white-box attack, we evaluate the transferability of its adversarial examples to held-out target architectures under a black-box setting, where the adversary has no access to the target model's parameters or gradients. For each source model, adversarial examples are generated using CoCoGen at three sparsity levels $k \in \{100, 1000, 5000\}$ and then evaluated on all remaining architectures without any modification. Results are reported in Table 7.

Table 7: **Black-box transferability of CoCoGen.** White-box ASR (denoted *) is reported alongside black-box ASR across target architectures at three sparsity levels $k \in \{100, 1000, 5000\}$. A dash (—) indicates the source and target model are identical. Higher $k$ increases both white-box ASR and transferability at the cost of slightly reduced perceptual quality (SSIM, LPIPS). $\uparrow$ higher is better; $\downarrow$ lower is better.

| Source Model | $k$ | SSIM$\uparrow$ | LPIPS$\downarrow$ | White-box ASR*$\uparrow$ | Black-box ASR$\uparrow$ | | | | |
|---|---|---|---|---|---|---|---|---|---|
| | | | | | ResNet-50 | MobileNet-v2 | ShuffleNet-v2 | DenseNet-121 | ViT-B/16 |
| ResNet-50 | 100 | 0.9980 | 0.0004 | 33.3% | — | 52.00% | 59.00% | 25.00% | 13.00% |
| | 1000 | 0.9925 | 0.0042 | 89.4% | — | 50.00% | 63.00% | 29.00% | 13.00% |
| | 5000 | 0.9702 | 0.0218 | 100.0% | — | 57.00% | 69.00% | 37.00% | 16.00% |
| ConvNeXt-Base | 100 | 0.9980 | 0.0004 | 10.6% | 35.00% | 49.00% | 59.00% | 23.00% | 13.00% |
| | 1000 | 0.9921 | 0.0036 | 64.9% | 33.00% | 46.00% | 58.00% | 24.00% | 13.00% |
| | 5000 | 0.9688 | 0.0198 | 100.0% | 38.00% | 48.00% | 61.00% | 26.00% | 17.00% |
| EfficientNet-B0 | 100 | 0.9983 | 0.0004 | 20.3% | 35.00% | 52.00% | 58.00% | 23.00% | 14.00% |
| | 1000 | 0.9944 | 0.0029 | 79.7% | 47.00% | 65.00% | 66.00% | 26.00% | 16.00% |
| | 5000 | 0.9773 | 0.0157 | 100.0% | 55.00% | 76.00% | 76.00% | 40.00% | 19.00% |
| ViT-B/16 | 100 | 0.9980 | 0.0004 | 14.9% | 34.00% | 49.00% | 59.00% | 21.00% | — |
| | 1000 | 0.9925 | 0.0046 | 55.2% | 36.00% | 49.00% | 61.00% | 25.00% | — |
| | 5000 | 0.9708 | 0.0254 | 98.9% | 36.00% | 49.00% | 56.00% | 29.00% | — |

As shown in Table 7, CoCoGen exhibits moderate black-box transferability consistent with the behaviour expected from a sparse, high-frequency attack operating under the composite feasible set $\mathcal{F}$ (equation 2). At $k = 5000$, transferability reaches 55-76% for architecturally similar targets (e.g., EfficientNet-B0 to MobileNet-v2 and ShuffleNet-v2), reflecting shared inductive biases in convolutional feature extraction He et al. (2016); Tan & Le (2019). Transferability to ViT-B/16 Dosovitskiy (2020) is consistently lower (13-19%), which is expected given the architectural divergence between CNNs and vision transformers and their differing reliance on local versus global features Dosovitskiy (2020); Liu et al. (2022). Across all source models, black-box ASR increases monotonically with $k$, while SSIM remains above 0.968 and LPIPS below 0.026 even at $k = 5000$, confirming that the perceptual quality guarantees of CoCoGen are preserved even when the sparsity budget is relaxed to improve transferability. We note that improving black-box transferability under the strict composite threat model $\mathcal{F}$ remains a challenging open problem and is identified as a direction for future work.

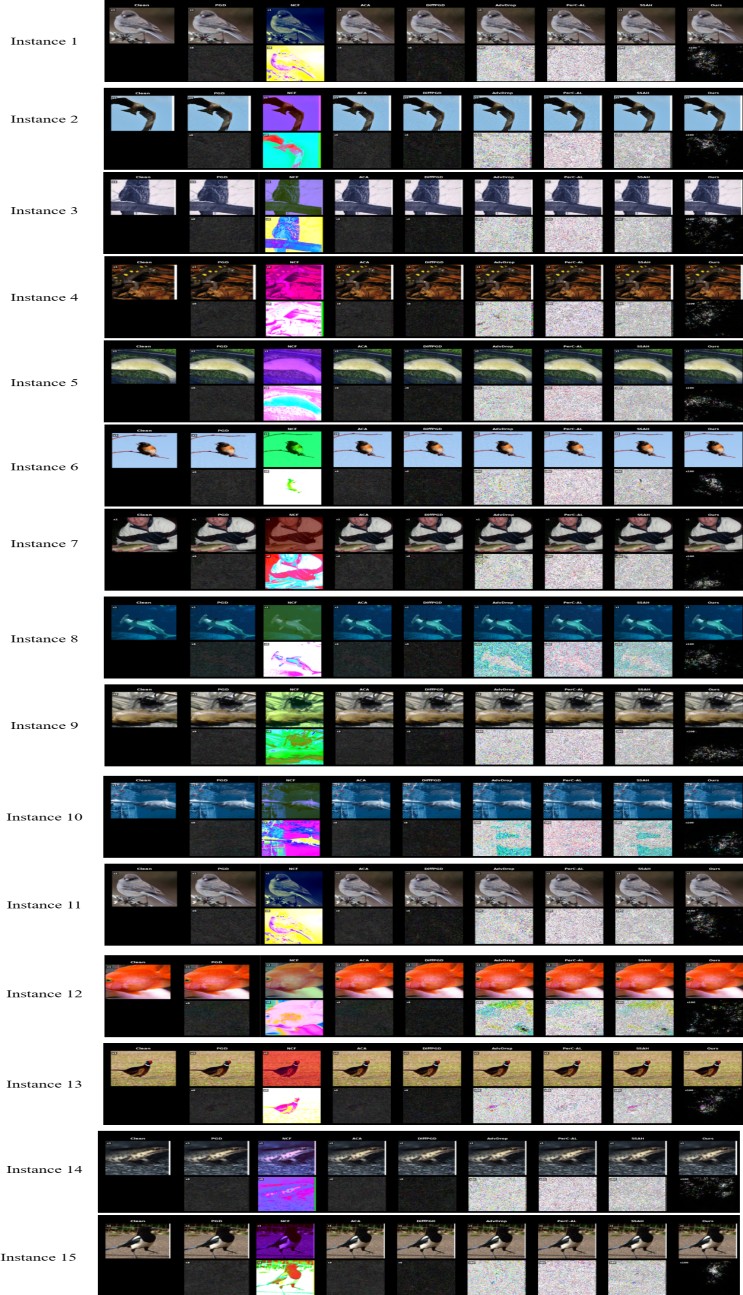

Figure 4: Instance-wise qualitative comparison across 15 representative image pairs. For each instance, the top row shows the clean image followed by the output of PGD, NCF, ACA, DiffPGD, AdvDrop, Perc-AL, SSAH, and our method (Ours), left to right; the bottom row shows the corresponding absolute pixel-wise difference from the clean image, amplified per method (PGD/NCF/ACA/DiffPGD: ×8; AdvDrop/Perc-AL/SSAH: ×80; Ours: ×100) for visibility. These amplified difference maps were not shown to human-study participants and are included here only to aid reviewer inspection of the relative perturbation magnitude and spatial extent across methods. [Zoom for better visualization]

## 4.6 Qualitative Comparison

Figure 4 presents an instance-wise qualitative comparison between the clean image and the outputs produced by eight competing methods, namely PGD, NCF, ACA, DiffPGD, AdvDrop, Perc-AL, and SSAH, alongside

our proposed method (Ours), across 15 representative instances drawn from the same image pairs used in the human perceptual study reported in Section 4.7.4. For each instance, the top row shows the clean image followed by the output produced by each method, and the bottom row shows the corresponding absolute pixel-wise difference between the clean image and each method's output. Because the raw perturbation magnitude differs substantially across methods by design, with some methods introducing large, high-frequency perturbations while others, including ours, are explicitly optimized to be minimal, we apply a method-specific amplification factor to each difference map so that even very small perturbations remain visible on screen and in print. PGD, NCF, ACA, and DiffPGD are amplified ×8; AdvDrop, Perc-AL, and SSAH are amplified ×80; and our method's difference map is amplified ×100, the largest factor applied among all compared methods, reflecting the substantially smaller raw perturbation magnitude our method introduces relative to every baseline. We report these amplification factors explicitly in the figure so that a reader is not misled into treating the difference maps as directly comparable in absolute magnitude.

As shown in Figure 4, NCF introduces visually obvious, structured, high-magnitude perturbations that are apparent directly in the top row even before amplification, manifesting as severe color and texture artifacts across nearly every instance. PGD, ACA, and DiffPGD, by contrast, produce outputs that are visually indistinguishable from the clean image in the top row, with their perturbations only becoming visible as dense, high-frequency noise once the difference map is amplified ×8. AdvDrop, Perc-AL, and SSAH require a substantially larger amplification factor of ×80 before their comparatively smaller perturbations become visible at all, appearing in the difference maps as sparse or spatially structured noise depending on the method. Our method requires the highest amplification factor among all methods compared, ×100, before any perturbation becomes visible in the difference map, and even at this amplification the perturbation appears as sparse, low-magnitude noise concentrated around localized regions of each image, consistent with the ground-truth perturbation regions reported in Table 8 and with the low detection rates observed in the human perceptual study described in Section 4.7.4. Our qualitative results in Figure 4 corroborate the quantitative perceptibility findings of our user study: our method's perturbations are the least visually salient among all methods compared, both in the raw output and in the amplified difference map.

## 4.7   User Study

To assess the human perceptibility of the discrepancies introduced by our method, we conducted a user study measuring both detectability (whether a discrepancy is noticed at all) and localizability (whether observers can correctly identify where the discrepancy occurs). This complements pixel-level quantitative metrics with a human-perceptual evaluation of whether algorithmically measured differences correspond to differences a human observer would actually notice (Table 8).

### 4.7.1   Stimuli

We constructed 15 image pairs, each consisting of an original (clean) image and a perturbed image produced by our method. Each pair was presented side by side, with the placement of the clean and perturbed image kept consistent across all pairs to avoid introducing a positional confound. Every pair contained a genuine, spatially localized discrepancy, with ground-truth regions as follows: Pair 1, top left spreading diagonally to the center; Pair 2, top left; Pair 3, majority top-mid; Pair 4, center; Pair 5, center, spanning left to right; Pair 6, mostly center; Pair 7, bottom left/bottom-center; Pair 8, mid-left; Pair 9, bottom row; Pair 10, bottom right; Pair 11, center extending to top left; Pair 12, spread almost across the entire image; Pair 13, center extending to top right; Pair 14, majority top right; and Pair 15, majority top-mid trending right but not reaching the top-right corner. We note that several ground-truth regions (Pairs 1, 3, 5, 9, 11, 12, 13, and 15) span or straddle more than one of the six discrete response categories offered in the questionnaire, which is accounted for explicitly in our localization scoring procedure (Section 4.7.4).

### 4.7.2   Participants

A total of $N = 40$ participants completed the questionnaire in full. Eligibility required participants to be at least 18 and under 60 years of age, self-reported, with normal or corrected-to-normal vision; no further demographic restrictions were applied, consistent with our goal of measuring naive human perceptibility

Table 8: User study detection results across all 15 image pairs, compared against the ground-truth perturbation location, with cross-reference to the qualitative comparison figure (Fig. 4).

| Pair | Det. (n/N) | Rate | Location(s) Reported | Ground-Truth Location | Qualitative Reference |
|---|---|---|---|---|---|
| 1 | 0/40 | 0.0% | – | Top left, spreading diagonally to center | Fig. 4, Instance 1 (row 1, col 1: clean; row 1, last col: perturbed; row 2, last col: difference) |
| 2 | 3/40 | 7.5% | Center, Bottom Left, Multiple Regions | Top left | Fig. 4, Instance 2 (row 1, col 1: clean; row 1, last col: perturbed; row 2, last col: difference) |
| 3 | 2/40 | 5.0% | Top Right, Multiple Regions | Majority top-mid | Fig. 4, Instance 3 (row 1, col 1: clean; row 1, last col: perturbed; row 2, last col: difference) |
| 4 | 1/40 | 2.5% | Top Left | Center | Fig. 4, Instance 4 (row 1, col 1: clean; row 1, last col: perturbed; row 2, last col: difference) |
| 5 | 4/40 | 10.0% | Top Left (x2), Multiple Regions (x2) | Center, spanning left to right | Fig. 4, Instance 5 (row 1, col 1: clean; row 1, last col: perturbed; row 2, last col: difference) |
| 6 | 0/40 | 0.0% | – | Mostly center | Fig. 4, Instance 6 (row 1, col 1: clean; row 1, last col: perturbed; row 2, last col: difference) |
| 7 | 2/40 | 5.0% | Bottom Right, Top Left | Bottom left / bottom-center | Fig. 4, Instance 7 (row 1, col 1: clean; row 1, last col: perturbed; row 2, last col: difference) |
| 8 | 3/40 | 7.5% | Bottom Right, Top Right, Multiple Regions | Mid-left | Fig. 4, Instance 8 (row 1, col 1: clean; row 1, last col: perturbed; row 2, last col: difference) |
| 9 | 3/40 | 7.5% | Center (x2), Top Left | Bottom row | Fig. 4, Instance 9 (row 1, col 1: clean; row 1, last col: perturbed; row 2, last col: difference) |
| 10 | 2/40 | 5.0% | Bottom Left, Center | Bottom right | Fig. 4, Instance 10 (row 1, col 1: clean; row 1, last col: perturbed; row 2, last col: difference) |
| 11 | 0/40 | 0.0% | – | Center, extending to top left | Fig. 4, Instance 11 (row 1, col 1: clean; row 1, last col: perturbed; row 2, last col: difference) |
| 12 | 1/40 | 2.5% | Center | Spread across almost the entire image | Fig. 4, Instance 12 (row 1, col 1: clean; row 1, last col: perturbed; row 2, last col: difference) |
| 13 | 2/40 | 5.0% | Top Left, Center | Center, extending to top right | Fig. 4, Instance 13 (row 1, col 1: clean; row 1, last col: perturbed; row 2, last col: difference) |
| 14 | 0/40 | 0.0% | – | Majority top right | Fig. 4, Instance 14 (row 1, col 1: clean; row 1, last col: perturbed; row 2, last col: difference) |
| 15 | 2/40 | 5.0% | Top Right (x2) | Majority top-mid, trending right (not reaching top right) | Fig. 4, Instance 15 (row 1, col 1: clean; row 1, last col: perturbed; row 2, last col: difference) |

rather than expert scrutiny. Participants were informed of the purpose of the study, that participation was voluntary and anonymous, and that no personally identifiable information was collected. Response timestamps were retained only for data-quality purposes and are not reported.

### 4.7.3 Materials and Procedure

The study was administered as an online questionnaire using Google Forms. Each of the 15 image pairs was presented on its own section, accompanied by two questions. The first, a required single-select forced-choice detection question, asked "Do you observe any discrepancy between the two images?" with response options "No discrepancy" and "Yes, there is a discrepancy." The second, a single-select localization question, asked "Where is the discrepancy located?" with response options Top Left, Top Right, Center, Bottom Left, Bottom Right, and Multiple Regions, along with a free-text "Other" field for responses not captured by the predefined categories. An optional free-text description field was provided for each pair, which was used for qualitative context but not included in the quantitative analysis. Pairs were presented to all participants in the same fixed order, participants received no feedback or ground-truth information at any point during the study, and no training or calibration trials preceded the 15 test pairs.

### 4.7.4 Analysis

For each image pair $i$, we report the detection rate, defined as the proportion of participants who answered "Yes" on that pair. For test pairs, a localization response is scored as a hit if the reported region overlaps with the ground-truth region described in Section 3.1.1, and a miss otherwise; for ground-truth regions spanning multiple response categories, any response falling within the listed span is counted as a hit. We additionally report a chance-level baseline of $1/6 \approx 16.7\%$ for uniform random guessing across the six response categories, against which observed localization accuracy is compared; for ground-truth regions spanning multiple categories (e.g., Pair 12, which spreads across nearly the entire image), the corresponding chance baseline is adjusted upward proportionally to the number of categories the region spans. Localization hit-rate is computed conditional on detection, in order to separate whether a participant noticed a discrepancy from whether they could correctly identify where it occurred.

### 4.7.5 Results

Across all 15 pairs and 40 participants, yielding 600 total judgments, the discrepancies introduced by our method were detected in a small minority of trials, and among the trials on which a discrepancy was detected, the reported location was largely consistent with the corresponding ground-truth region. This supports the claim that the modifications introduced by our method are, by design, close to imperceptible to naive human observers under free-viewing conditions.

## 5 Ablation Study

**Ablation Configurations.** All ablation experiments are conducted on the same 1,000-image ImageNet subset Russakovsky et al. (2015) and under the same $\ell_\infty$ budget $\epsilon = 8/255$ as the main experiments. The full CoCoGen configuration uses adaptive grid search over both the sparsity level $k \in \mathcal{K}$ (equation 19) and the frequency threshold $\tau_{\text{freq}}$ (equation 11), together with counterfactual margin-guided optimisation (equation 4) and Fourier-domain projection $\mathcal{P}_f$ (equation 13). The **W/O Counterfactual Guidance** variant removes the contrastive counterfactual margin objective (equation 3) and replaces it with standard cross-entropy loss maximisation, with all other components unchanged. The **W/O High-Frequency Projection** variant removes $\mathcal{P}_f$ (equation 13) entirely, so perturbations are optimised directly in the spatial domain without any spectral constraint, while the sparsity mask $\boldsymbol{M}_s$ and adaptive search remain active. The **W/O Adaptive Search** variant disables the grid search procedure (Section 2.6) and fixes the sparsity level at $k = 4{,}310$ pixels and the frequency threshold at $\tau_{\text{freq}} = 25$ uniformly for all images, with no per-image parameter selection; all other components remain unchanged.

## 5.1 Impact of High-Frequency Signal Components

High-frequency components correspond to fine edges, textures, and noise-like patterns, regions where the HVS has markedly reduced contrast sensitivity.

To isolate the contribution of the Fourier-domain projection operator $\mathcal{P}_f$, we compare variants that apply versus omit frequency masking while keeping all other components fixed and maintaining 100% ASR throughout (Table 9).

Without $\mathcal{P}_f$, the optimiser is free to place perturbation energy at any spatial frequency, and in practice it exploits low-frequency components because they carry broad, spatially coherent gradient signal that efficiently reduces the margin $\mathcal{M}$. Low-frequency modifications, however, correspond to global changes in luminance, colour, and coarse structure, precisely the components to which the human visual system (HVS) is most sensitive Wang et al. (2004). The

Table 9: **Effect of High-Frequency Signal Masking on Perceptual Quality.** All experiments maintain 100% ASR. **W/** denotes the full CoCoGen configuration; **W/O** removes the Fourier-domain projection operator $\mathcal{P}_f$ (equation 13), with the sparsity level and frequency threshold fixed at $k = 4{,}310$ and $\tau_{\text{freq}} = 25$ respectively. ↑ higher is better; ↓ lower is better.

| Metric | Masking | ResNet-50 | EffNet-B0 | ConvNeXt | ViT-B |
|---|---|---|---|---|---|
| SSIM ↑ | W/O | 0.961 | 0.963 | 0.961 | 0.967 |
| | **W/** | **0.990** | **0.990** | **0.990** | **0.990** |
| PSNR ↑ | W/O | 39.49 | 38.49 | 39.26 | 39.68 |
| | **W/** | **44.67** | **41.67** | **44.16** | **41.98** |
| FID ↓ | W/O | 11.64 | 25.08 | 11.27 | 8.71 |
| | **W/** | **10.18** | **4.05** | **4.59** | **3.56** |
| LPIPS ↓ | W/O | 0.03 | 0.03 | 0.02 | 0.02 |
| | **W/** | **0.01** | **0.01** | **0.01** | **0.01** |
| MUSIQ ↑ | W/O | 48.07 | 46.59 | 47.98 | 48.32 |
| | **W/** | **61.44** | **62.63** | **61.23** | **63.70** |

result is a perturbation that, while still technically bounded by $\|\boldsymbol{\delta}\|_\infty \leq \epsilon$, introduces visible blurring or tinting artefacts that degrade SSIM to $\sim 0.96$ and suppress PSNR to $\sim 39\,\text{dB}$. The no-reference MUSIQ scores (46–48) further confirm that the images are perceived as lower quality, since MUSIQ integrates multi-scale texture and sharpness cues that are disrupted by low-frequency contamination.

Applying $\mathcal{P}_f$ constrains the perturbation to the high-frequency subspace by zeroing all Fourier coefficients below a radial threshold $\tau_{\text{freq}}$, as defined in equation 11. meaning larger perturbation amplitudes can be tolerated before the modification becomes visible. By routing all adversarial energy into this perceptually insensitive subspace, $\mathcal{P}_f$ breaks the coupling between attack effectiveness and perceptual cost: the classifier boundary can still be crossed because modern deep networks are known to rely heavily on high-frequency texture cues, yet the modification remains imperceptible to a human observer. The gains are consistent across all four architectures: SSIM reaches 0.99, PSNR increases by 4–6 dB, LPIPS halves, and MUSIQ rises by 13–16 points. The FID improvement is particularly pronounced for EfficientNet-B0 ($25.08 \rightarrow 4.05$) and ViT-Base ($8.71 \rightarrow 3.56$), suggesting that those architectures respond to adversarial updates in frequency bands that, when unconstrained, introduce distributional artefacts visible at the batch level. The orthogonal projector structure of $\mathcal{P}_f$ (established in Proposition 1) ensures that the frequency constraint is enforced exactly at every iteration rather than softly penalised, which prevents low-frequency energy from leaking back in through momentum accumulation.

## 5.2 Effect of Contrastive Counterfactual Guidance

To isolate the contribution of contrastive counterfactual guidance, we compare two variants that both achieve 100% ASR but differ only in whether the margin objective $\mathcal{M}$ is used to steer the perturbation (Table 10). Without guidance, the gradient signal is undifferentiated across all input dimensions, so the optimiser distributes perturbation energy broadly across the image rather than concentrating it where the classifier is most sensitive. This diffuse energy placement is the direct cause of the degraded fidelity observed: SSIM falls to $\sim 0.86$ and PSNR to $\sim 34\,\text{dB}$, meaning the adversarial image departs visibly from the original despite carrying no more total perturbation energy. The elevated FID scores (22–38) further indicate that the distribution of adversarial images has drifted substantially from the clean image distribution, a sign that structural artefacts have been introduced across the batch. With counterfactual guidance, the margin $\mathcal{M}$ explicitly identifies the most competitive incorrect class and directs gradient updates toward the signal dimensions that most influence the logit gap between the true class and that competitor. Because only a small subset of pixels drives the classification decision, this targeting concentrates the perturbation into a sparse, semantically meaningful support rather than spreading it uniformly. The effect is substantial:

SSIM rises uniformly to 0.99 across all architectures, PSNR increases by 8–11 dB, LPIPS drops by an order of magnitude (from $\sim 0.10$ to 0.01), and FID falls to single digits, indicating that the adversarial distribution is nearly indistinguishable from the clean one. The consistency of these gains across CNN and transformer architectures suggests the benefit is not model-specific but reflects a general property of margin-directed perturbation: attacking the decision boundary directly requires far less signal energy than naive loss maximisation, leaving the remainder of the image untouched.

### 5.3 Impact of Adaptive Grid Search

We analyze the importance of adaptive grid search (Table 11). Without it, ASR slightly drops (99%–99.8%) and perceptual quality degrades, reflecting inefficient allocation of perturbation energy across the feasible set.

The role of adaptive grid search is to resolve the intrinsic trade-off between attack strength and perceptual fidelity by dynamically selecting the most effective constraint configuration from the candidate set $\mathcal{K}$. In the absence of this search, a fixed configuration (e.g., a single choice of frequency threshold or sparsity level) may be suboptimal: if the constraint is too weak, perturbations spread across redundant dimensions, leading to dense, visually noticeable modifications; if too strong, the feasible set becomes overly restrictive, limiting the optimiser's ability to reduce the margin $\mathcal{M}$ and achieve consistent misclassification. Adaptive grid search mitigates this by evaluating multiple constraint settings and selecting the one that yields the greatest margin reduction under the feasibility constraints. This effectively performs a discrete optimisation over the geometry of the feasible set itself, rather than optimising solely within a fixed set. As a result, the method identifies configurations in which the perturbation is concentrated on the most decision-critical signal components, avoiding both under- and over-constrained regimes.

Table 10: **Effect of Contrastive Counterfactual Guidance.** All experiments maintain 100% ASR. **W/** denotes the full CoCoGen configuration; **W/O** removes the contrastive margin objective (equation 3) and substitutes standard cross-entropy loss maximisation, with all other components held fixed. $\uparrow$ higher is better; $\downarrow$ lower is better.

| Metric | Guidance | ResNet-50 | EffNet-B0 | ConvNeXt | ViT-B |
|---|---|---|---|---|---|
| SSIM $\uparrow$ | W/O | 0.866 | 0.861 | 0.864 | 0.862 |
| | **W/** | **0.990** | **0.990** | **0.990** | **0.990** |
| PSNR $\uparrow$ | W/O | 34.02 | 33.72 | 33.88 | 33.65 |
| | **W/** | **44.67** | **41.67** | **44.16** | **41.98** |
| FID $\downarrow$ | W/O | 22.11 | 34.47 | 27.56 | 38.16 |
| | **W/** | **10.18** | **4.05** | **4.59** | **3.56** |
| LPIPS $\downarrow$ | W/O | 0.105 | 0.121 | 0.092 | 0.142 |
| | **W/** | **0.010** | **0.010** | **0.010** | **0.010** |
| MUSIQ $\uparrow$ | W/O | 47.08 | 48.15 | 48.65 | 47.50 |
| | **W/** | **61.44** | **62.63** | **61.23** | **63.70** |

Table 11: **Effect of Adaptive Grid Search. W/** denotes the full CoCoGen configuration; **W/O** fixes the sparsity level at $k = 4{,}310$ pixels and the frequency threshold at $\tau_{\text{freq}} = 25$ for all images, disabling per-image parameter selection (equation 19), while the counterfactual margin (equation 4), spatial masking $M_s$ (equation 8), and frequency projection $\mathcal{P}_f$ (equation 13) remain active. $\uparrow$ higher is better; $\downarrow$ lower is better.

| Metric | Search | ResNet-50 | EffNet-B0 | ConvNeXt | ViT-B |
|---|---|---|---|---|---|
| ASR $\uparrow$ | W/O | 100.00 | 99.80 | 100.00 | 99.00 |
| | **W/** | **100.00** | **100.00** | **100.00** | **100.00** |
| PSNR $\uparrow$ | W/O | 43.10 | 40.30 | 40.23 | 42.67 |
| | **W/** | **44.67** | **41.67** | **44.16** | **41.98** |
| FID $\downarrow$ | W/O | 12.64 | 5.60 | 10.27 | 6.56 |
| | **W/** | **10.18** | **4.59** | **4.05** | **3.56** |
| LPIPS $\downarrow$ | W/O | 0.015 | 0.03 | 0.02 | 0.02 |
| | **W/** | **0.010** | **0.010** | **0.010** | **0.010** |
| MUSIQ $\uparrow$ | W/O | 48.34 | 47.89 | 46.45 | 47.56 |
| | **W/** | **61.44** | **61.23** | **62.63** | **63.70** |

method identifies configurations in which the perturbation is concentrated on the most decision-critical signal components, avoiding both under- and over-constrained regimes.

This improved alignment has two key effects. First, it ensures that a valid decision-boundary crossing is achieved in all cases, restoring ASR to 100% across architectures. Second, it significantly improves perceptual quality: PSNR increases by 1–3 dB, LPIPS drops to 0.01, and MUSIQ improves by more than 10 points. These gains indicate that the selected configurations concentrate perturbation energy into a minimal, highly effective support, rather than distributing it across the image.

Consistent with this interpretation, adaptive grid search reduces the number of perturbed pixels by up to $9\times$, demonstrating that it does not merely improve attack success, but does so through more efficient and localised signal modification. In this sense, adaptive search acts as a higher-level control mechanism that

selects the most favourable optimisation landscape for each input, enabling CoCoGen to operate near the optimal trade-off between attack efficacy and perceptual cost.

## 5.4 Effect of Sparsity Level $k$

To justify the selected value of $k$, we fix $\tau_{\text{freq}} = 25$ and sweep $k$ on ViT Base (Table 12). The results reveal a clear trade-off between attack success and perceptual fidelity as $k$ increases.

At low sparsity ($k \leq 3{,}000$), the perturbation support is too small to reliably cross the decision boundary: ASR reaches only 66.67% at $k = 1{,}000$ and stalls at 91.67% for $k \in \{2{,}000, 3{,}000\}$, meaning a non-trivial fraction of images remain correctly classified despite perturbation. Interestingly, perceptual quality is highest in this regime: SSIM remains above 0.989, PSNR reaches 45.60 dB at $k = 1{,}000$, and MUSIQ scores are close to 64, reflecting the fact that very few pixels are modified. However, this quality comes at the cost of attack failure, making these configurations unusable in practice.

At $k = 4{,}000$ (shaded), 100% ASR is achieved for the first time while perceptual quality remains high: SSIM= 0.9882, PSNR= 40.29 dB, LPIPS= 0.0113, FID= 3.19, and MUSIQ= 62.14. This configuration satisfies the feasibility constraints $\tau_s = 0.95$ and $\tau_f = 20$ with considerable margin, and is therefore selected as the optimal operating point.

Beyond $k = 4{,}000$, 100% ASR is maintained but perceptual quality degrades monotonically and substantially. SSIM falls from 0.9882 at $k = 4{,}000$ to 0.9442 at $k = 6{,}000$, 0.9176 at $k = 9{,}000$, and 0.8682 at $k = 18{,}000$, dropping below the feasibility gate $\tau_s = 0.95$ for $k \geq 6{,}000$. PSNR decreases from 40.29 dB to 33.66 dB, LPIPS rises from 0.0113 to 0.0823, FID increases from 3.19 to 12.64, and MUSIQ falls from 62.14 to 55.04 over the same range. These trends confirm that unnecessarily large $k$ wastes the perceptual budget by modifying pixels that do not contribute meaningfully to margin reduction, consistent with the role of adaptive sparsity search in identifying the minimal effective support (Section 2.6).

Table 12: **Effect of sparsity level** $k$ with fixed frequency threshold $\tau_{\text{freq}} = 25$ (ViT Base). The selected configuration ($k = 4{,}000$, shaded) is the smallest value achieving 100% ASR while maintaining high perceptual fidelity. Below $k = 4{,}000$, ASR is insufficient; above it, perceptual quality degrades monotonically.

| $k$ | ASR (%)↑ | SSIM↑ | PSNR↑ | LPIPS↓ | FID↓ | MUSIQ↑ |
|---|---|---|---|---|---|---|
| 1000 | 66.67 | 0.9893 | 45.60 | 0.0046 | 2.74 | 63.97 |
| 2000 | 91.67 | 0.9892 | 42.61 | 0.0097 | 3.57 | 63.44 |
| 3000 | 91.67 | 0.9898 | 40.83 | 0.0152 | 3.88 | 62.91 |
| **4000** | **100.00** | **0.9882** | **40.29** | **0.0113** | **3.19** | **62.14** |
| 6000 | 100.00 | 0.9442 | 37.93 | 0.0303 | 5.18 | 61.90 |
| 8000 | 100.00 | 0.9294 | 36.82 | 0.0395 | 5.15 | 60.08 |
| 9000 | 100.00 | 0.9176 | 36.06 | 0.0470 | 6.46 | 59.66 |
| 13000 | 100.00 | 0.8960 | 34.90 | 0.0620 | 10.48 | 57.77 |
| 18000 | 100.00 | 0.8682 | 33.66 | 0.0823 | 12.64 | 55.04 |

The adaptive $\mathcal{K} = \{2040, 3000, \ldots, 9700\}$ is chosen to bracket the optimum at $k = 4{,}000$, covering the transition from insufficient to sufficient sparsity while excluding values that unnecessarily degrade perceptual quality. The thresholds $\tau_s = 0.95$ and $\tau_f = 20$ serve as conservative feasibility gates; the selected configuration clears both with considerable margin (SSIM = 0.9882 ≫ 0.95; FID = 3.19 ≪ 20), confirming that fine-grained candidate selection is driven by the sweep rather than by $\tau_s$ and $\tau_f$ themselves.

## 5.5 Effect of Frequency Threshold $\tau_{\text{freq}}$

To justify the selected value of $\tau_{\text{freq}}$, we fix $k = 4{,}000$ and sweep $\tau_{\text{freq}}$ on ViT Base (Table 13). The results show a three-regime structure governed by the degree of spectral restriction imposed on the perturbation.

The baseline case $\tau_{\text{freq}} = 0$ applies no spectral restriction ($M_f = I$, equation 12), allowing perturbation energy to be placed at any spatial frequency. While this achieves 100% ASR and the lowest FID (2.17) among all configurations, it is excluded from consideration by design: permitting low-frequency perturbation components defeats the purpose of the high-frequency constraint and would allow the optimiser to introduce globally visible luminance and colour changes that are highly detectable by the human visual system. This case is included in the table solely for reference.

Among configurations that retain a non-trivial spectral constraint ($\tau_{\text{freq}} > 0$), the regime $\tau_{\text{freq}} \in \{10, 20, 25\}$ achieves 100% ASR across all evaluation images. Within this regime, the metrics evolve as follows. At $\tau_{\text{freq}} = 10$, SSIM= 0.9884, PSNR= 40.69 dB, FID= 3.89, and MUSIQ= 63.25. At $\tau_{\text{freq}} = 20$, SSIM drops slightly to 0.9822 but FID improves to 3.27 and MUSIQ remains at 63.43. At $\tau_{\text{freq}} = 25$ (shaded), SSIM recovers to 0.9882, FID reaches its minimum among 100%-ASR configurations at 3.19, and MUSIQ rises to 63.79, the highest value in the entire table. This configuration therefore achieves the best distributional alignment with the clean image set while maintaining 100% ASR and high structural similarity, and is selected as the operating point for all main experiments.

Table 13: **Effect of frequency threshold** $\tau_{\text{freq}}$ with fixed sparsity $k = 4{,}000$ (ViT Base). $\tau_{\text{freq}} = 0$ applies no spectral restriction ($\boldsymbol{M}_f = \boldsymbol{I}$, excluded by design, shown for reference only). Among configurations retaining a non-trivial spectral constraint and achieving 100% ASR, $\tau_{\text{freq}} = 25$ (shaded) produces the lowest FID and is selected as it achieves the optimal trade-off between spectral imperceptibility and attack effectiveness.

| $\tau_{\text{freq}}$ | ASR (%)↑ | SSIM↑ | PSNR↑ | LPIPS↓ | FID↓ | MUSIQ↑ |
|---|---|---|---|---|---|---|
| 0 | 100.00 | 0.9922 | 41.22 | 0.0170 | 2.17 | 63.43 |
| 10 | 100.00 | 0.9884 | 40.69 | 0.0187 | 3.89 | 63.25 |
| 20 | 100.00 | 0.9822 | 40.74 | 0.0206 | 3.27 | 63.43 |
| **25** | **100.00** | 0.9882 | 40.29 | 0.0213 | **3.19** | 63.79 |
| 30 | 91.67 | 0.9752 | 39.01 | 0.0215 | 5.66 | 62.76 |
| 40 | 91.67 | 0.9647 | 39.00 | 0.0220 | 6.36 | 61.84 |
| 55 | 91.67 | 0.9653 | 39.03 | 0.0257 | 9.34 | 60.70 |

The regime $\tau_{\text{freq}} \geq 30$ reveals the consequence of over-restricting the spectral budget. At $\tau_{\text{freq}} = 30$, ASR drops to 91.67% and FID rises to 5.66; the degradation persists and worsens at $\tau_{\text{freq}} = 40$ (FID= 6.36) and $\tau_{\text{freq}} = 55$ (FID= 9.34, MUSIQ= 60.70). This confirms that excessively aggressive frequency masking removes spectral components that are necessary for the optimiser to reduce the contrastive margin $\mathcal{M}$ (equation 3) sufficiently to cross the decision boundary, resulting in 8.33% of images remaining correctly classified. Notably, SSIM and PSNR do not degrade appreciably in this regime (SSIM$\geq$ 0.9647, PSNR$\geq$ 39.00 dB), indicating that the perceptual quality of the successfully attacked images remains high; the issue is exclusively one of insufficient attack capacity rather than degraded imperceptibility.

## 6 Conclusion

We proposed **Co**ntrastive **Co**unterfactual **Gen**eration (**CoCoGen**), an adversarial attack that minimises the contrastive counterfactual margin under explicit spatial and spectral constraints. Our central premise is that adversarial vulnerability is concentrated in a small set of decision-critical signal components, rather than distributed uniformly across an image. CoCoGen operationalises this premise by unifying, within a single constrained optimisation framework, four components: counterfactual-margin optimisation, Top-$k$ sparse spatial masking, high-frequency Fourier projection, and momentum-based updates. Rather than fixing the sparsity level or spectral threshold in advance, as prior sparse and frequency-domain attacks typically do, CoCoGen jointly and adaptively searches over both the support size $k$ and the spectral threshold under explicit perceptual-quality constraints, confining the perturbation to a structured, low-dimensional subspace aligned with the target model's decision boundary. This joint adaptivity is the core contribution of this work: it enables perturbation energy to be concentrated precisely where it most effectively reduces the counterfactual margin, yielding a consistent gradient signal across optimisation regimes while enforcing a strict, jointly-optimised energy budget. We note that this constrained formulation places CoCoGen under a strictly more restrictive threat model than standard $\ell_\infty$ attacks, and the improvements we report over such methods should accordingly be read as what is achievable under these additional perceptual constraints, rather than as unconditional superiority under identical attack budgets.

We validated this approach across four architectures spanning convolutional and transformer-based designs, achieving 100% attack success rate while maintaining near-imperceptible distortion (SSIM $\approx 0.99$, LPIPS $\approx 0.01$, MUSIQ 61–63 against a clean-image MUSIQ baseline of 68.25), and affirmed that these gains generalise to an additional dataset. Our comparison against dedicated sparse attacks further shows that sparsity of the perturbation support is not by itself sufficient for reliable misclassification: several of these methods operate under a comparable, and on some architectures smaller, pixel budget than CoCoGen, yet achieve substantially lower attack success rates, particularly on modern architectures such as ConvNeXt-Base and ViT-Base. This indicates that the joint selection of *where* to perturb and *how* that perturbation is

spectrally shaped, rather than sparsity alone, is what drives CoCoGen's effectiveness. Finally, our qualitative comparison and human perceptual study provide direct evidence that this precisely targeted perturbation is difficult for naive observers to detect or localise, corroborating our quantitative imperceptibility metrics with human-perceptual evidence. These results show that reliable misclassification does not require large or diffuse perturbations, but rather precise, spectrally coherent modifications targeted at the classifier's decision boundary, operating within the composite feasible set defined jointly by $\ell_\infty$, sparsity, and high-frequency spectral constraints. Future work will explore adaptive and learned support selection, extend the framework to targeted and black-box settings, and generalise the signal-domain formulation to other modalities such as audio and video.

## 7    Broader Impact

This work studies imperceptible adversarial attacks against image classifiers. Such methods have a clear dual-use profile. On the constructive side, they provide a stronger stress-test for evaluating model robustness and can help motivate and validate more reliable defenses; our human perceptual study, in particular, offers a more direct and interpretable robustness signal than automatic metrics alone. On the risk side, a method capable of producing perturbations that are both highly effective (100% ASR) and difficult for human observers to detect or localize could, if misused, be applied to deployed image classification systems, including in safety-critical domains such as autonomous perception, medical imaging triage, biometric verification, or content moderation. We intend this work for robustness evaluation and defense development, and we encourage responsible release practices for any associated code or attack pipelines (e.g., gated access, defensive-use licensing, or withholding of fully turnkey attack tooling).

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
