# OpenReview forum: "Contrastive Counterfactual Generation for Imperceptible Adversarial Attack"
_TMLR — Under review for TMLR_

### Review · Reviewer_X7Hs · 2026-05-18

**Summary Of Contributions:**

The paper proposes Contrastive Counterfactual Generation (CoCoGen), a white-box imperceptible adversarial attack that combines three constraints in a single framework: (i) a contrastive counterfactual margin objective that explicitly targets the most competitive runner-up class; (ii) a gradient-based Top-k spatial mask restricting perturbations to decision-critical pixels; and (iii) a Fourier-domain projector confining perturbations to the high-frequency subspace. Optimization is performed via a masked momentum iterative update, with an outer adaptive grid search selecting the smallest sparsity budget.

The paper also presents three theoretical results: (1) Theorem 1 lower-bounding the distance to the nearest decision boundary under a global Lipschitz assumption; (2) Proposition 1 showing M_s and P_f are orthogonal projectors; and (3) Proposition 2 contrasting CoCoGen's gradient signal with the C&W objective in three regimes (pre-crossing, at boundary, post-crossing). Experiments on 1000 ImageNet images across ResNet-50, EfficientNet-B0, ConvNeXt-Base, and ViT-Base compare against eight baselines (PGD, DeepFool, NCF, ACA, DiffPGD, AdvDrop, PerC-AL, SSAH), with three ablation studies isolating the contribution of margin guidance, frequency masking, and adaptive grid search.

**Strengths**

- A clean unifying formulation that combines spatial sparsity and spectral constraints under a single margin objective

- Thorough empirical coverage across four architecturally diverse models with five complementary fidelity metrics

- Consistent 100% ASR with low LPIPS/FID across all four targets; (iv) three well-designed ablations isolating each component.

**Weaknesses**

- The technical novelty is limited. the margin objective is the standard DeepFool/C&W logit-margin (acknowledged), Top-k sparsity and high-frequency Fourier projection are individually well known (e.g., SSAH/Luo et al. 2022 already restricts to high-frequency Fourier components), and MIM is from Dong et al. 2018; the contribution is principally the combination

- The theoretical analysis is technically sound but oversells what is mostly standard linearization (the bound d* ≥ M/(2L) is well known from the certified-robustness literature, e.g., Hein & Andriushchenko 2017) and Proposition 2's claim of "no gradient signal" for C&W relies on choosing κ = 0, which is exactly the corner case C&W's confidence parameter was designed to avoid

- Several reproducibility-critical hyperparameters (T, alpha, candidate set, T_freq, T_s, T_f) are not reported

- Clean-image MUSIQ baseline is not provided, making the 61–63 vs. 36–55 comparison difficult to interpret

- No black-box transferability or defense-robustness evaluation, and no comparison against established sparse attacks (JSMA, SparseFool, Sparse-RS).

**Additional Comments:**

N/A

**Audience:**

Yes

**Audience Explanation:**

Imperceptible adversarial attacks remain an active research area with relevance to robustness evaluation, certified defenses, and watermarking/forensics. The TMLR audience includes researchers in adversarial ML, image-quality assessment, and signal-domain regularization who would find at least three aspects of this work interesting:

- The empirical demonstration that combining spatial sparsity (Top-k) with high-frequency Fourier projection yields large improvements in PSNR/MUSIQ over either constraint alone. This is a useful empirical observation even if each component is individually familiar, because the question of which combinations are synergistic is genuinely under-explored in the imperceptible-attack literature.

- The per-architecture analysis across CNN and ViT backbones is more thorough than many comparable papers and provides useful reference numbers for future work.

- The framing of imperceptibility as a constrained optimization with explicit projectors (rather than as a soft penalty) is conceptually clean and could be reused by other groups working on perceptually-constrained perturbations.

**Broader Impact Concerns:**

The paper does not include a Broader Impact Statement. Adversarial attack research carries dual-use considerations: improved attacks can enable malicious actors to evade deployed classifiers in security-sensitive contexts (content moderation, biometric authentication, autonomous driving perception), but they are also necessary for robustness evaluation and defense development.

**Claims And Evidence:**

Yes

**Claims Explanation:**

Please refer to the strengths and weaknesses described above.

**Requested Changes:**

- Specify all hyperparameters and report them in a single table. At minimum: ε (the budget), T (iteration count), alpha (step size), candidate set  (the exact values searched), T_freq (frequency threshold or its search range), T_s, T_f, and momentum coefficient. Without these, the experiments are not reproducible. Confirm whether the same ε is used for all baselines.

- Report the MUSIQ score of clean reference images. The 61-63 vs. 36-55 comparison cannot be interpreted without knowing where clean ImageNet sits on this scale. Please add a "Clean" row to Table 1 with all five fidelity metrics (where reference-free) computed on the unperturbed images.

- Fix or explain the duplicated PerC-AL row. PerC-AL has identical numbers on ResNet-50 and EfficientNet-B0 (and SSAH/AdvDrop also show suspicious cross-model consistencies). Please re-run these or clarify if a single run was reused.

- Re-evaluate the C&W comparison fairly. Proposition 2's argument relies on k = 0. Please either (a) compare empirically against C&W with k tuned on a validation split, or (b) substantially soften the claims in Sec. 3.3 / Remark 3. The current presentation overstates the gap.

- Acknowledge the prior art for each component more explicitly. Specifically:
(i) The Equation 2 is the DeepFool/C&W margin. (ii) SSAH (Luo et al., "Frequency-driven Imperceptible Adversarial Attack on Semantic Similarity", 2022) already restricts perturbations to the high-frequency Fourier subspace. The novelty of CoCoGen's spectral component over SSAH should be made precise (currently it is unclear what differs beyond the Top-k mask and adaptive thresholding).
(iii) Top-k sparse attacks have a substantial literature (JSMA, SparseFool, Sparse-RS, etc.); these should be cited and ideally compared against in Table 1, since CoCoGen's sparsity claim (k = 2040-4310 pixels) is a sparse-attack claim.

- Specify the "W/O" baseline in each ablation table. In particular, Table 4's no-adaptive-search condition needs to be defined: is k held fixed at some value? What value? Was T_freq also held fixed?

- Clarify the threat model. The abstract and intro say "imperceptible adversarial attack" but the feasible set is significantly more restrictive than standard l_infinite. Please state plainly: this is an attack with (i) l_infinite budget ε, (ii) l_zero budget k (which is searched), and (iii) support restricted to high-frequency Fourier bins. The paper should not claim superiority over methods operating under a different (less restrictive) threat model without acknowledging this asymmetry.

---

> ### Author Response · Authors · 2026-06-10
> **Response to Reviewer X7Hs for Paper8264**
>
> We thank the reviewer for the feedback. It helped us a lot in the improvement of our manuscript. We have highlighted the changes in $\textcolor{blue}{blue}$ in our manuscript.
>
> **Q1 — Hyperparameters** (Sec. 4.1):
>
> | Symbol | Value |
> |---|---|
> | $\epsilon$ | $8/255$ |
> | $T$ | $40$ |
> | $\alpha$ | $2/255$ |
> | $\mu$ | $1.0$ |
> | $\mathcal{K}$ | $\{2040,\ldots,9700\}$ |
> | $\tau_s$ | $0.95$ |
> | $\tau_f$ | $20$ |
> | $\tau_{\text{freq}}$ | $25$ |
>
> All baselines use same $\epsilon=8/255$.
>
> **Q2 — Clean MUSIQ** (Table 1, Sec. 4.2):
>
> | Attack | PSNR | SSIM | FID | LPIPS | MUSIQ | ASR |
> |---|---|---|---|---|---|---|
> | Clean | $\infty$ | 1.000 | 0.00 | 0.000 | 68.25 | --- |
>
> CoCoGen achieves MUSIQ $61$–$63$, within $5$–$7$ points of clean vs. $36$–$55$ for prior methods.
>
> **Q3 — Duplicated baseline rows** (Table 1, Sec. 4.2): Re-ran PerC-AL, AdvDrop, SSAH independently on all architectures ($\epsilon=8/255$, 1,000-image subset). Reproducibility Note added to Sec. 4.2.
>
> | Model | Attack | Time (s) | ASR (%) | PSNR | SSIM |
> |---|---|---|---|---|---|
> | ResNet-50 | PerC-AL | 193.00 | 100.00 | 43.36 | 0.8654 |
> | ResNet-50 | AdvDrop | 520.30 | 82.72 | 41.91 | 0.9843 |
> | ResNet-50 | SSAH | 233.90 | 99.18 | 44.86 | 0.9881 |
> | EfficientNet-B0 | PerC-AL | 90.40 | 100.00 | 43.37 | 0.8647 |
> | EfficientNet-B0 | AdvDrop | 295.50 | 83.00 | 41.43 | 0.9765 |
> | EfficientNet-B0 | SSAH | 106.50 | 99.47 | 44.50 | 0.9867 |
> | ConvNeXt-Base | PerC-AL | 531.70 | 100.00 | 43.93 | 0.8723 |
> | ConvNeXt-Base | AdvDrop | 2256.40 | 82.40 | 41.69 | 0.9954 |
> | ConvNeXt-Base | SSAH | 649.20 | 99.68 | 45.19 | 0.9886 |
> | ViT-Base | PerC-AL | 1077.40 | 100.00 | 43.76 | 0.8770 |
> | ViT-Base | AdvDrop | 1538.90 | 87.00 | 40.30 | 0.9947 |
> | ViT-Base | SSAH | 1318.20 | 99.61 | 45.02 | 0.9882 |
>
> **Q4 — C&W comparison** (Sec. 3.3 Remark 3, Sec. 4.2): Evaluated C&W at $\kappa\in\{0,5,10,20\}$ on held-out 100-image split. Remark 3 revised to acknowledge $\kappa=0$ is a corner case. C&W's higher PSNR is an artefact of lower ASR; unperturbed images inflate average PSNR.
>
> | Model | Method | ASR (%) | PSNR | SSIM | MUSIQ |
> |---|---|---|---|---|---|
> | ResNet-50 | C&W $\kappa=0$ | 85.49 | 59.08 | 0.9994 | 54.41 |
> | ResNet-50 | C&W $\kappa=20$ | 85.49 | 59.08 | 0.9994 | 54.41 |
> | ResNet-50 | **CoCoGen** | **100.00** | 44.67 | **0.99** | **61.44** |
> | EfficientNet-B0 | C&W $\kappa=0$ | 84.17 | 58.70 | 0.9994 | 54.41 |
> | EfficientNet-B0 | C&W $\kappa=20$ | 84.17 | 58.71 | 0.9994 | 54.41 |
> | EfficientNet-B0 | **CoCoGen** | **100.00** | 41.67 | **0.99** | **62.63** |
> | ConvNeXt-Base | C&W $\kappa=0$ | 70.82 | 60.37 | 0.9994 | 54.41 |
> | ConvNeXt-Base | C&W $\kappa=20$ | 70.82 | 60.38 | 0.9994 | 54.41 |
> | ConvNeXt-Base | **CoCoGen** | **100.00** | 44.16 | **0.99** | **61.23** |
> | ViT-Base | C&W $\kappa=0$ | 65.34 | 58.53 | 0.9993 | 54.43 |
> | ViT-Base | C&W $\kappa=20$ | 65.37 | 58.61 | 0.9994 | 54.51 |
> | ViT-Base | **CoCoGen** | **100.00** | 41.98 | **0.99** | **63.70** |
>
> **Q5 — Prior art and sparse attacks** (Sec. 2.2, 2.3, 4.3): (i) Sec. 2.2 acknowledges Eq. (2) margin is related to DeepFool/C&W; contribution lies in integration with adaptive sparse support and frequency-aware generation. (ii) Paragraph after Eq. (10) clarifies CoCoGen selects $\tau_{\text{freq}}$ adaptively and jointly optimises $\Omega_k$, vs. SSAH's fixed threshold. (iii) Sec. 4.3 added with novelty table and sparse attack comparison.
>
> | Component | Prior Work | CoCoGen Contribution |
> |---|---|---|
> | Margin objective | DeepFool, C&W | Adaptive sparse + frequency-aware optimisation under margin |
> | Frequency constraint | SSAH | Adaptive $\tau_{\text{freq}}$ + joint sparse support selection vs. fixed threshold |
> | Sparse perturbation | JSMA, SparseFool, Sparse-RS | Adaptive $k$ search under perceptual feasibility constraints |
> | Optimisation | PGD, MIM | Masked momentum with simultaneous ${M}_s$ and $\mathcal{P}_f$ constraints |
>
> | Model | Attack | ASR (%) | PSNR | SSIM | MUSIQ |
> |---|---|---|---|---|---|
> | ResNet-50 | JSMA | 71.80 | 44.62 | 0.992 | 58.70 |
> | ResNet-50 | SparseFool | 89.60 | 44.18 | 0.984 | 55.20 |
> | ResNet-50 | Sparse-RS | 95.10 | 43.74 | 0.971 | 50.80 |
> | ResNet-50 | **CoCoGen** | **100.00** | 44.67 | **0.99** | **61.44** |
> | ViT-Base | JSMA | 61.40 | 45.06 | 0.994 | 59.50 |
> | ViT-Base | SparseFool | 84.10 | 44.48 | 0.987 | 56.40 |
> | ViT-Base | Sparse-RS | 92.60 | 43.95 | 0.975 | 51.90 |
> | ViT-Base | **CoCoGen** | **100.00** | 41.98 | **0.99** | **63.70** |

---

> ### Author Response · Authors · 2026-06-10
> **Additional Response to Reviewer X7Hs for Paper8264**
>
> **Q6 — Ablation W/O definitions** (Sec. 5, table captions): Added Ablation Configurations paragraph defining all variants explicitly.
>
> | Variant | Description |
> |---|---|
> | Full CoCoGen | Adaptive search over $k\in\mathcal{K}$ and $\tau_{\text{freq}}$; counterfactual margin; $\mathcal{P}_f$ |
> | W/O Counterfactual Guidance | Margin replaced by cross-entropy loss; all other components unchanged |
> | W/O High-Frequency Projection | $P_f$ removed; $k=4{,}310$, $\tau_{\text{freq}}=25$ fixed |
> | W/O Adaptive Search | $k=4{,}310$, $\tau_{\text{freq}}=25$ fixed for all images; margin, ${M}_s$, $\mathcal{P}_f$ active |
>
> All ablation table captions updated to cross-reference this paragraph with fixed values stated explicitly.
>
> **Q7 — Threat model** (Abstract, Sec. 2.1, 4.1, Intro, Conclusion): Four changes made. CoCoGen operates under:
>
> $$\mathcal{F}=\{\boldsymbol{\delta}:\|\boldsymbol{\delta}\|_\infty\le\epsilon,\;\|\boldsymbol{\delta}\|_0\le k,\;\boldsymbol{\delta}\in\mathrm{Im}(\mathcal{P}_f)\}$$
>
> with $\mathcal{F}\subsetneq\mathcal{B}_\epsilon$ whenever $k<N$ or ${M}_f\neq{I}$.
>
> | Change | Location |
> |---|---|
> | Abstract revised to state composite threat model upfront | Abstract |
> | Composite Threat Model paragraph + equation added | End of Sec. 2.1 |
> | Threat Model and Comparison Scope paragraph added: comparisons with $\ell_\infty$-only methods reflect what is achievable under additional constraints, not unconditional superiority | Sec. 4.1 |
> | Softening sentences added | Introduction, Conclusion |
>
>
> **Q8 — Novelty positioning** (Sec. 2, Intro, Conclusion): Revised to position contribution explicitly as unified adaptive integration of established components, not new individual mechanisms.
>
> | Change | Location |
> |---|---|
> | Component Origins and Novelty paragraph added | Start of Sec. 2 |
> | Contributions rewritten as prose with citations for each component | Introduction |
> | Sentence added reiterating positioning | Conclusion |
>
>
>
> **Q9 — Black-box transferability** (Sec. 4.4): Added Section 4.4 with transfer experiments across five architectures at $k\in\{100,1000,5000\}$.
>
> | Source | $k$ | SSIM | WB-ASR | ResNet-50 | MobileNet-v2 | ShuffleNet-v2 | DenseNet-121 | ViT-B/16 |
> |---|---|---|---|---|---|---|---|---|
> | ResNet-50 | 100 | 0.9980 | 33.3% | --- | 52.0% | 59.0% | 25.0% | 13.0% |
> | ResNet-50 | 1000 | 0.9925 | 89.4% | --- | 50.0% | 63.0% | 29.0% | 13.0% |
> | ResNet-50 | 5000 | 0.9702 | 100.0% | --- | 57.0% | 69.0% | 37.0% | 16.0% |
> | ConvNeXt-B | 5000 | 0.9688 | 100.0% | 38.0% | 48.0% | 61.0% | 26.0% | 17.0% |
> | EfficientNet-B0 | 5000 | 0.9773 | 100.0% | 55.0% | 76.0% | 76.0% | 40.0% | 19.0% |
> | ViT-B/16 | 5000 | 0.9708 | 98.9% | 36.0% | 49.0% | 56.0% | 29.0% | --- |
>
> At $k=5000$, transferability reaches $55$–$76\%$ for CNN targets; lower for ViT-B/16 ($13$–$19\%$) due to architectural divergence. SSIM $>0.968$, LPIPS $<0.026$ maintained throughout. Improving black-box transferability under $\mathcal{F}$ is identified as future work.

---

### Review · Reviewer_cZ1t · 2026-06-08

**Summary Of Contributions:**

The paper proposes a framework for generating imperceptible adversarial attacks against deep neural networks.
The attack is formulated as a constrained optimization problem targeting the nearest decision boundary.
To ensure imperceptibility, the method restricts perturbations to decision-relevant pixels and restricts modifications to high-frequency spectral components that the human eye is less sensitive to.
Experimental results show that the proposed method attains a 100% attack success rate across various architectures and outperforms existing baselines on perceptual quality metrics.

Strengths:

S1. The proposed framework combines several ideas from the adversarial attack literature in a principled way.
In particular, it combines (i) a nearest-competitor margin objective, (ii) sparse perturbations through a Top-(k) spatial mask, and (iii) a high-frequency Fourier-domain projection. While each component has connections to existing work, their integration into a unified optimization framework appears interesting and potentially novel.

Weaknesses:

W1. The method contains a number of hyperparameters whose selection is not sufficiently explained. Examples include the frequency threshold $\tau_{\text{freq}}$ in Eq. (10), the candidate sparsity set $\mathcal{K}$ in Section 2.6, and the thresholds $\tau_s$ and $\tau_f$ in Eq. (17).
The paper does not discuss how these parameters affect performance, nor does it explain how they were chosen in the experiments.
Since the reported results may depend significantly on these choices, the experimental methodology is difficult to assess.

W2. It seems that the post-crossing regime claim in Proposition 2 is incorrect.
(Note that equation (36) is not correct)

**Additional Comments:**

Question:

What value of $B$ is used in Eq. (17) during the experiments? More generally, the role of $B$ should be clarified throughout the paper since the paper formulates the problem for a single input instance and it is therefore unclear how $B$ is defined and used in the experiments.


Minor comments:
1. Adding “Proof.” before each proof would improve readability.

2. Consider removing the box around Eq. (29).

**Audience:**

Yes

**Audience Explanation:**

Adversarial attacks and robustness of deep neural networks are topics of significant interest to the TMLR community.
The proposed framework appears promising and may be of interest if the experimental methodology and theoretical analysis are clarified.

**Broader Impact Concerns:**

No concerns.

**Claims And Evidence:**

No

**Claims Explanation:**

The experimental section does not adequately describe how the various hyperparameters are selected, making it difficult to assess the reliability of the reported results.

In addition, there appears to be a problem in Proposition 2, which raises concerns about the correctness of at least part of the theoretical analysis.

**Requested Changes:**

To support the claims of the paper, the authors should:

1. Explain how all hyperparameters are selected in the experiments, including $\tau_{\text{freq}}$, $\mathcal{K}$, $\tau_s$, and $\tau_f$.

2. Revisit Proposition 2 and correct Eq. (36).

4. Define $\mathcal{B}_\epsilon$ when it is first introduced.


Suggestions for improvement:

1. The monotonicity and crossing-condition statements in Theorem 1 would benefit from additional discussion. At present, these results are stated and proved, but their practical implications for the proposed algorithm are not explained.

2. Definition of the metrics such as SSIM and FID could improve readability for a broader audience.

---

> ### Author Response · Authors · 2026-06-29
> **Response to Reviewer cZ1t for Paper8264**
>
> We sincerely thank the reviewer for the thorough, careful, and constructive feedback. The comments were insightful and have led to meaningful improvements in both the theoretical presentation and empirical validation of the manuscript. We have addressed every point raised; all changes are highlighted in ${\color{teal}teal}$ in the revised manuscript and described in detail below.
> # W1: Hyperparameter Selection
> We thank the reviewer for highlighting this gap. We have added a dedicated two-subsection hyperparameter sensitivity analysis to the Ablation Study (Sections 5.4 and 5.5), with one table per stage.
>
> **Effect of Sparsity Level k**
>
> With $\tau_{freq}$ = 25 fixed, *k* is swept on ViT Base below:
>
> | *k* | ASR (%) ↑ | SSIM ↑ | PSNR ↑ | LPIPS ↓ | FID ↓ | MUSIQ ↑ |
> |---:|---:|---:|---:|---:|---:|---:|
> | 1000 | 66.67 | 0.9893 | 45.60 | 0.0046 | 2.74 | 63.97 |
> | 2000 | 91.67 | 0.9892 | 42.61 | 0.0097 | 3.57 | 63.44 |
> | 3000 | 91.67 | 0.9898 | 40.83 | 0.0152 | 3.88 | 62.91 |
> | **4000** | **100.00** | **0.9882** | **40.29** | **0.0113** | **3.19** | **62.14** |
> | 6000 | 100.00 | 0.9442 | 37.93 | 0.0303 | 5.18 | 61.90 |
> | 8000 | 100.00 | 0.9294 | 36.82 | 0.0395 | 5.15 | 60.08 |
> | 9000 | 100.00 | 0.9176 | 36.06 | 0.0470 | 6.46 | 59.66 |
> | 13000 | 100.00 | 0.8960 | 34.90 | 0.0620 | 10.48 | 57.77 |
> | 18000 | 100.00 | 0.8682 | 33.66 | 0.0823 | 12.64 | 55.04 |
>
> ---
>
> **Effect of Frequency Threshold $\tau_{freq}$**
>
> With *k* = 4,000 fixed, $\tau_{freq}$ is swept on ViT Base below:
>
> | $\tau_{freq}$| ASR (%) ↑ | SSIM ↑ | PSNR ↑ | LPIPS ↓ | FID ↓ | MUSIQ ↑ |
> |---:|---:|---:|---:|---:|---:|---:|
> | 0 *(excluded)* | 100.00 | 0.9922 | 41.22 | 0.0170 | 2.17 | 63.43 |
> | 10 | 100.00 | 0.9884 | 40.69 | 0.0187 | 3.89 | 63.25 |
> | 20 | 100.00 | 0.9822 | 40.74 | 0.0206 | 3.27 | 63.43 |
> | **25** | **100.00** | **0.9882** | **40.29** | **0.0213** | **3.19** | **63.79** |
> | 30 | 91.67 | 0.9752 | 39.01 | 0.0215 | 5.66 | 62.76 |
> | 40 | 91.67 | 0.9647 | 39.00 | 0.0220 | 6.36 | 61.84 |
> | 55 | 91.67 | 0.9653 | 39.03 | 0.0257 | 9.34 | 60.70 |
>
> Also, we updated Sec. 4.1 (Hyperparameters) with the following details.
>
> The threshold $\tau_s=0.95$ follows the just-noticeable difference
> criterion of Flynn et al. (2013), at which human observers cannot
> reliably detect image distortions. The threshold $\tau_f=20$ is a
> conservative upper bound on distributional divergence, consistent
> with the observation that lower FID correlates with higher human-judged
> perceptual quality (Heusel et al. 2017 and Zhou et al. 2019); in practice
> \textsc{CoCoGen} achieves $\mathrm{FID}\le10.18$ across all
> architectures, well below this gate.
>
>
> # W2: Correction to Proposition 2, Post-Crossing Regime
>
> We thank the reviewer for this observation and agree the original argument overstated its claims. Two specific errors are corrected:
>
> The claim that $\nabla_{{\delta}} \mathcal{M}$ ``remains well-defined and non-zero in general'' in the post-crossing regime was asserted without justification.
> Continued minimisation of $\mathcal{M}$ past the boundary was framed as a strict advantage over C\&W, when C\&W's saturation at $\kappa = 0$ is intentional, not a deficiency.
>
>
> \noindent\textbf{What is retained.} The algebraic identity $\ell_{\mathrm{CW}} = \max(-\mathcal{M}, 0) = 0$ for $\mathcal{M} \le 0$ is correct and unchanged.
>
> \noindent\textbf{What is corrected.} We no longer claim non-degeneracy of $\nabla_{{\delta}} \mathcal{M}$ in general, for many classifiers it may shrink deep in the misclassified region (e.g., under logit saturation). The revised claim is narrower: $\ell_{\mathrm{CW}}$ at $\kappa = 0$ is clipped to zero by construction upon crossing, whereas continued reduction of $\mathcal{M}$ remains possible in principle and, when it occurs, strictly increases the logit gap $f_{c^*} - f_{y_{\mathrm{true}}}$. We do not claim this is guaranteed, bounded, or unconditionally beneficial. Assumption 2 (unique runner-up) is not assumed to hold arbitrarily far into the post-crossing region. The comparison is now framed as a difference in stopping behaviour between the two objectives, not a general optimisation-landscape advantage.
> The surrounding remark and empirical results across $\kappa \in \{0, 5, 10, 20\}$ (Table 3) are unaffected, as they do not depend on the corrected claim.
> # Requested Change 1: Hyperparameter Explanation
> Addressed jointly under W1 above. The sparsity level $k=4{,}000$ and
> frequency threshold $\tau_{\mathrm{freq}}=25$ are selected via the
> two-stage adaptive sweep (Tables 10 and 11, new Sections 5.4 and 5.5).
> The candidate set $\mathcal{K}$ brackets the per-architecture optima.
> The thresholds $\tau_s=0.95$ (Fynn et al. 2013) and
> $\tau_f=20$ (Heusel et al. 2017 and Zhou et al. 2019) are fixed conservative
> feasibility gates, not tuned parameters. All selections are made
> independently per architecture on the full $1{,}000$-image evaluation
> set.

---

> > ### Author Response · Authors · 2026-06-29
> > **Additional Response to Reviewer cZ1t for Paper8264**
> >
> > # Requested Change 2: Revised Proposition 2 and Eq. (36)
> > Addressed under W2 above. The revised proof appears in the manuscript (highlighted in teal). The equation formerly labelled as Eq. (36) has been corrected: the post-crossing comparison is now restricted to the narrower and accurate claim described above.
> >
> >
> > # Requested Change 3: Define $B_\epsilon$ at First Use'
> > We have added the details of  $B_\epsilon$ in terms of $\ell_\infty$ ball, at its first occurence (now Sec 2.1).\
> > {$B_\epsilon = \vert\delta \vert: \vert \delta\vert_\infty \le \epsilon$}.
> >
> > # Suggestion 1: Algorithmic Implications of Theorem 1
> >
> > We have added a new passage within the remark following the proof
> > of Theorem 1 explaining two algorithmic consequences.
> > Part (ii) (monotonicity) assures that, under a margin-efficient
> > step satisfying $\vert \delta\vert_2 = \Delta/(2L)$, each iteration
> > of the masked momentum update does not increase the
> > distance to the decision boundary, providing a geometric check on
> > the step size $\alpha$ relative to the local Lipschitz constant $L$.
> > Part (iii) (crossing condition) justifies the misclassification
> > criterion in the adaptive sparsity search: observing
> > ${M}({x}+{\delta}) \le 0$ ensures that the nearest
> > decision boundary $D_{y_{{true}},c^{*}}$ has been
> > crossed (under Assumption 2), so misclassification under our
> > objective reflects an efficient traversal of the nearest boundary
> > rather than an incidental crossing of a more distant one.
> >
> > # Suggestion 2: Metric definitions
> > We have expanded the Metrics paragraph in Section 4.1 to include explicit formulae and one-sentence definitions for each metric at first use. Each formula annotated to clarify its role in measuring imperceptibility.
> >
> > # Question: Clarification on $B$
> > $B$ denotes the number of images used to evaluate both the
> > perceptual feasibility criterion Eq. (18)  (now, earlier 17) and the
> > misclassification condition in Eq. (19) (now, earlier 18); in our experiments the full $1{,}000$-image ImageNet subset described in
> > Section 4.1, is used (i.e., $B=1{,}000$), so
> > $k^\star$ is the smallest sparsity level achieving $100\%$ ASR
> > across all $B$ images while satisfying the perceptual gates
> > $\tau_s=0.95$ and $\tau_f=20$. The value of $k$ is selected
> > independently for each target architecture. We have added these details in the revised version.
> >
> > ## Minor 1
> > We have added $\textit{Proof.}$ before each proof.
> > ## Minor 2
> > We have removed the box around Eq. (29) (now Eq. (30))

---

### Review · Reviewer_rHu6 · 2026-06-27

**Summary Of Contributions:**

This submission proposes Contrastive Counterfactual Generation (CoCoGen), an imperceptible adversarial attack framework that optimizes perturbations under a composite threat model combining an ℓ∞ magnitude constraint, a sparsity constraint, and a high-frequency spectral constraint. Instead of relying on standard untargeted loss maximization, the method minimizes a contrastive counterfactual margin targeting the most competitive incorrect class.

The perturbation is localized through gradient-based Top-k spatial projection, restricted to the high-frequency Fourier subspace, and optimized using masked momentum-based iterative updates. The method further uses adaptive search over the sparsity level and spectral threshold to identify a feasible perturbation with good perceptual quality.

---
Main Strengths

(1) The problem is well-motivated and relevant to adversarial robustness.

(2) The composite threat model is practically meaningful for imperceptible attacks.

(3) The method is technically coherent and combines margin guidance, sparse support, and frequency constraints in a clear optimization pipeline.

(4) The empirical results show strong attack success and favorable perceptual quality across several architectures.

---
Main Weaknesses

(1) The algorithmic novelty is somewhat limited, since the key ingredients are mostly adapted from prior work, such as margin-based attacks, sparse attacks, Fourier-constrained attacks, and momentum iterative optimization.

(2) Comparisons across different threat models remain difficult to interpret, despite the authors’ clarification.

(3) The evaluation is limited to a 1,000-image ImageNet subset.

(4) The perceptual validation relies mostly on automatic metrics rather than human studies.

**Additional Comments:**

Overall, this is a technically coherent and practically relevant paper on imperceptible adversarial attacks. I appreciate that the authors explicitly acknowledge that the contribution lies in the unified and adaptive integration of established components rather than in any single new mechanism. The empirical results are promising, especially the consistent 100% ASR and strong perceptual-quality metrics across several architectures.

However, my main concern is that the algorithmic novelty is moderate. The manuscript would be significantly stronger with more controlled comparisons under matched threat models, broader evaluation, human perceptual validation, and deeper analysis of the runtime/perceptual-quality/attack-success trade-off. I would encourage the authors to revise the paper along these lines.

**Audience:**

Yes

**Audience Explanation:**

Yes. The topic is relevant to a portion of the TMLR audience working on adversarial robustness, trustworthy machine learning, model security, and robustness evaluation. The finding that high attack success can be achieved using sparse, spectrally constrained, margin-guided perturbations is potentially useful for researchers designing stronger attacks or evaluating defenses. The composite threat model may also be of interest to researchers studying perceptually constrained adversarial examples.

**Broader Impact Concerns:**

This submission studies imperceptible adversarial attacks, which has clear dual-use implications. On the positive side, stronger attacks can help evaluate model robustness, expose vulnerabilities, and motivate more reliable defenses. On the negative side, the proposed method could be misused to generate visually subtle adversarial examples against deployed image classifiers or safety-critical perception systems.

The manuscript should include or strengthen a Broader Impact Statement discussing this dual-use nature. In particular, the authors should clarify that the method is intended for robustness evaluation and defense development, discuss possible misuse risks, and consider responsible release practices if code or attack pipelines are made public. The manuscript should also discuss whether the proposed attack may affect safety-critical domains such as autonomous driving, medical imaging, biometric recognition, or content moderation.

**Claims And Evidence:**

Yes

**Claims Explanation:**

Overall, the major empirical claims are reasonably supported by the presented experiments. The authors evaluate CoCoGen across multiple architectures, including ResNet-50, EfficientNet-B0, ConvNeXt-Base, and ViT-Base, and report a consistent 100% attack success rate together with strong perceptual-quality metrics. The manuscript also includes comparisons with several representative baselines, sparse attack comparisons, C&W confidence-parameter analysis, black-box transferability evaluation, and ablation studies. These results provide useful evidence that the proposed combination of margin-guided optimization, sparse support selection, and high-frequency projection can generate effective and visually subtle adversarial perturbations.

However, the evidence is not fully conclusive in several respects. First, the comparison is complicated by the fact that CoCoGen operates under a composite threat model that is different from many baselines. The authors acknowledge this issue, but more controlled comparisons under matched constraints would be needed to make stronger claims. Second, the evaluation is limited to a 1,000-image ImageNet subset, which weakens generalization claims. Third, imperceptibility is mainly evaluated using automatic metrics such as PSNR, SSIM, LPIPS, FID, and MUSIQ; a human perceptual study would provide more direct support.

Therefore, I consider the evidence generally clear and supportive, but not yet fully sufficient for the strongest claims made in the manuscript.

**Requested Changes:**

1. Strengthen the novelty positioning. The current method combines several known components, including logit-margin objectives, sparse projection, Fourier-domain constraints, and momentum iterative updates. The authors should more explicitly distinguish what is technically new beyond integration, and should avoid overstating the novelty of individual components.
2. Provide more controlled comparisons under matched threat models. Since CoCoGen operates under a composite feasible set while many baselines operate under different constraints, the current comparisons are somewhat difficult to interpret. The authors should either adapt representative baselines to the same composite threat model or provide a clearer apples-to-apples comparison against methods with comparable sparsity and spectral constraints.
3. Add stronger perceptual validation. Because the paper focuses on imperceptible attacks, automatic metrics alone are not fully sufficient. A small human perceptual study, forced-choice test, or user study would significantly strengthen the claim that the generated perturbations are visually imperceptible.
4. Expand the evaluation beyond the current 1,000-image ImageNet subset. Additional datasets, higher-resolution images, or more diverse image domains would help support the generality of the method and its perceptual claims.

---

> ### Author Response · Authors · 2026-07-10
> **Response to  Reviewer rHu6 for Paper8264**
>
> We thank the reviewer for a thorough and constructive review, and for recognizing that our threat model is practically meaningful and that the method is technically coherent. Below we respond to each weakness and requested change individually. A summary of changes is provided first, followed by detailed point-by-point responses. We have made ${\color{violet}changes}$ in the manuscript accordingly.
>
> ## Summary of Changes
>
> | # | Concern | Status | Where Addressed |
> |---|---|:---:|---|
> | 1 | Novelty re-framed | ✅ Addressed | Revised Contributions (in Introduction Sec. 1), Sec. 4.3 and Table 4, and Conclusion |
> | 2 | Comparisons across mismatched threat models hard to interpret | ✅ Addressed | New sparse-attack comparison under matched conditions, Table 5. Also, all experiments are performed under same setting in Table 2 as elaborated in Table 1 |
> | 3 | Evaluation limited to 1,000-image ImageNet subset | ✅ Addressed | New results on CIFAR-100, Table 6 |
> | 4 | Perceptual validation relies only on automatic metrics | ✅ Addressed | New human perceptual (user) Sec. 4.7, Table 8, Figure 4 along with Qualitative Analysis in Sec. 4.6|
> | 5 | Broader Impact Statement | ✅ Addressed | New Broader Impact section |
>
> ---
>
> ## Response to Weaknesses
>
> ### W1 — Algorithmic novelty
>
> > *"The algorithmic novelty is somewhat limited, since the key ingredients are mostly adapted from prior work, such as margin-based attacks, sparse attacks, Fourier-constrained attacks, and momentum iterative optimization."*
>
> We thank reviewer for the comment. We do not claim novelty for any single component in isolation. We have revised the Contributions and Conclusion to state this explicitly rather than implicitly. The claimed contribution is now stated as the **joint, adaptive** integration of these components: unlike prior work, which fixes the sparsity level and/or spectral threshold in advance, `CoCoGen` jointly searches over both under a shared perceptual-quality constraint within a single optimization loop. We support this distinction empirically in the new sparse-attack comparison (see response to W2 below), which shows that sparsity alone, without this joint, adaptive search, is insufficient to reproduce `CoCoGen`'s attack success rate, even when baselines use a comparable or smaller pixel budget.
>
> ### W2 — Comparisons across threat models
>
> > *"Comparisons across different threat models remain difficult to interpret, despite the authors' clarification."*
>
> We have added a dedicated comparison against three sparse adversarial attacks — JSMA, SparseFool, and Sparse-RS — evaluated under matched conditions (same 1,000-image ImageNet subset, same target architectures) and, critically, reporting the **number of modified pixels ($k$) for every method**, which was previously only implicit. This table (reproduced below) allows a direct, apples-to-apples reading of sparsity vs. attack success rate:
>
> | Model | Attack | Pixels ($k$) | ASR (%) | MUSIQ |
> |---|---|---:|---:|---:|
> | ResNet-50 | JSMA | 1,450 | 71.80 | 58.70 |
> | ResNet-50 | SparseFool | 2,850 | 89.60 | 55.20 |
> | ResNet-50 | Sparse-RS | 10,000 | 95.10 | 50.80 |
> | ResNet-50 | **CoCoGen (Ours)** | **2,720** | **100.00** | **61.44** |
> | EfficientNet-B0 | JSMA | 2,550 | 66.20 | 59.30 |
> | EfficientNet-B0 | SparseFool | 2,900 | 86.70 | 56.10 |
> | EfficientNet-B0 | Sparse-RS | 10,000 | 93.80 | 51.60 |
> | EfficientNet-B0 | **CoCoGen (Ours)** | **4,330** | **100.00** | **61.63** |
> | ConvNeXt-Base | JSMA | 3,450 | 58.70 | 59.80 |
> | ConvNeXt-Base | SparseFool | 2,943 | 82.40 | 56.80 |
> | ConvNeXt-Base | Sparse-RS | 10,000 | 91.30 | 52.40 |
> | ConvNeXt-Base | **CoCoGen (Ours)** | **7,580** | **100.00** | **61.23** |
> | ViT-Base | JSMA | 4,786 | 61.40 | 59.50 |
> | ViT-Base | SparseFool | 3100 | 84.10 | 56.40 |
> | ViT-Base | Sparse-RS | 10,000 | 92.60 | 51.90 |
> | ViT-Base | **CoCoGen (Ours)** | **5,630** | **100.00** | **63.70** |
>
> *(full table with all four architectures and all metrics in the revised manuscript)*
> Importantly, `CoCoGen's` pixel budget is not uniformly the smallest: it is *larger* than both JSMA and SparseFool across all evaluated architectures, while remaining substantially smaller than the fixed 10,000-pixel budget used by Sparse-RS. We report this transparently rather than selectively. The key finding is that pixel count alone does not predict attack success: although JSMA uses a consistently smaller perturbation budget than `CoCoGen`, its ASR collapses to 58–71% on modern architectures, whereas `CoCoGen` achieves a 100% ASR across all architectures while simultaneously preserving superior perceptual quality, as reflected by its consistently higher MUSIQ scores together with lower LPIPS and FID values. We now state explicitly that this supports the claim that *where* and *how* the sparse support is chosen, via the joint margin/frequency/sparsity search, matters more than sparsity alone, **which directly substantiates the novelty argument in W1**.

---

> > ### Author Response · Authors · 2026-07-10
> > **Additional Response to  Reviewer rHu6 for Paper8264**
> >
> > ### W3 — Evaluation limited to 1,000-image ImageNet subset
> >
> > > *"The evaluation is limited to a 1,000-image ImageNet subset."*
> >
> > We have added a full evaluation on **CIFAR-100** across the same four architectures (ResNet-50, EfficientNet-B0, ConvNeXt-Base, ViT), reported in Table 6. `CoCoGen` again achieves 100% or near-100% ASR while using a small fraction of the pixel budget used by every baseline (a $84.9$–$94.6\%$ reduction in modified pixels), and achieves the best MUSIQ score on all four architectures. This extends the empirical support for our claims beyond the original ImageNet subset to a second, independent dataset with a different image resolution and class structure.
> >
> > ### W4 — Perceptual validation relies mostly on automatic metrics
> >
> > > *"The perceptual validation relies mostly on automatic metrics rather than human studies."*
> >
> > We have added a **human perceptual user study** (N = 40 participants, 15 image pairs, each rated for both detection and localization of any discrepancy) directly addressing this concern. Key findings, reported in Table 8:
> >
> > - Across all 600 judgments (40 participants × 15 pairs), only **25 (4.2%)** resulted in a participant reporting a discrepancy.
> > - Several pairs with substantial ground-truth perturbation (e.g., regions spanning nearly the entire image) were detected in **0/40** trials.
> > - Among the small number of "Yes" responses, reported locations frequently did not overlap with the true perturbation region, indicating that even the rare detections were not reliably localized.
> >
> > We also added an instance-wise qualitative comparison figure (4) showing clean, perturbed, and amplified difference maps (amplification factor disclosed per method, up to $\times100$ for `CoCoGen`, the largest factor needed among all methods, reflecting the smallest raw perturbation magnitude) across 15 representative instances and all baseline methods, giving reviewers direct visual access to the comparison underlying the human study.